# Quadrupolar $^{23}$Na$^+$ NMR relaxation as a probe of subpicosecond collective dynamics in aqueous electrolyte solutions

Iurii Chubak [1,7], Leeor Alon[2,3,7], Emilia V. Silletta [4,5], Guillaume Madelin [2,3], Alexej Jerschow [6] ✉ & Benjamin Rotenberg [1] ✉

Nuclear magnetic resonance relaxometry represents a powerful tool for extracting dynamic information. Yet, obtaining links to molecular motion is challenging for many ions that relax through the quadrupolar mechanism, which is mediated by electric field gradient fluctuations and lacks a detailed microscopic description. For sodium ions in aqueous electrolytes, we combine ab initio calculations to account for electron cloud effects with classical molecular dynamics to sample long-time fluctuations, and obtain relaxation rates in good agreement with experiments over broad concentration and temperature ranges. We demonstrate that quadrupolar nuclear relaxation is sensitive to subpicosecond dynamics not captured by previous models based on water reorientation or cluster rotation. While ions affect the overall water retardation, experimental trends are mainly explained by dynamics in the first two solvation shells of sodium, which contain mostly water. This work thus paves the way to the quantitative understanding of quadrupolar relaxation in electrolyte and bioelectrolyte systems.

The proper characterization and modeling of the solvation structure of alkaline cations (e.g., Li$^+$, Na$^+$, and K$^+$) in aqueous solution are of considerable interest both in physiological systems[1–8] and electrolytes used for electrochemical devices[9–13]. Nuclear magnetic resonance (NMR) spectroscopy provides an excellent source of dynamic and structural information for a number of nuclear species, including $^{23}$Na with a nuclear spin 3/2 and close to 100% natural abundance that produces the second strongest NMR signal after protons in biological tissues[14]. The NMR sensitivity of sodium is 9.2% of that of proton, while its typical concentration can be three, or more, orders of magnitude lower. Thus, in biological systems the sodium signal-to-noise ratio is 3000–12,000 times lower than that of $^1$H[4]. Nonetheless, the longitudinal relaxation time $T_1$ of $^{23}$Na (typically 40 ms and below) is short

compared to that of $^1$H (on the order of seconds)[4], allowing for rapid averaging of the signals such that quantitative analysis is made possible within reasonable time scales[15].

The shortness of the $^{23}$Na NMR relaxation times is due to a fluctuating quadrupole interaction related to the changes in the solvation shell and the proximity of other ions[16]. The relaxation rate is determined from a combination of the strength of the electric field gradient (EFG) at the nucleus quantified by means of the quadrupolar coupling constant (QCC) $C_Q$ and the characteristic correlation time $\tau_c$ with which the memory of fluctuations is lost. While the knowledge of $C_Q$ and $\tau_c$ can potentially provide information about the hydration sphere structure[17,18] and useful dynamic properties (e.g., diffusion coefficients, viscosity, or conductivity[19–21]), respectively, their

[1]Sorbonne Université CNRS, Physico-Chimie des électrolytes et Nanosystèmes Interfaciaux, F-75005 Paris, France. [2]New York University School of Medicine, Department of Radiology, Center for Biomedical Imaging, 660 First Avenue, New York, NY 10016, USA. [3]Center for Advanced Imaging Innovation and Research, Department of Radiology, New York University Grossman School of Medicine, New York, NY 10016, USA. [4]Universidad Nacional de Córdoba, Facultad de Matemática, Astronomía, Física y Computación, Medina Allende s/n, X5000HUA Córdoba, Argentina. [5]Instituto de Física Enrique Gaviola, CONICET, Medina Allende s/n, X5000HUA Córdoba, Argentina. [6]New York University, Department of Chemistry, 100 Washington Square E, New York, NY 10003, USA. [7]These authors contributed equally: Iurii Chubak, Leeor Alon. ✉e-mail: alexej.jerschow@nyu.edu; benjamin.rotenberg@sorbonne-universite.fr

unambiguous determination from the experimentally-measured rates in solution has remained essentially impossible[22].

Different models have been suggested to rationalize quadrupolar relaxation using dielectric description[23,24], mode-coupling analysis[25], definite molecular processes (e.g., water reorientation[26–28] and collective symmetry-breaking fluctuations[29]), or Brownian rotational diffusion[19–21]. Ab initio[30–34] and classical[29,35–42] molecular dynamics (MD) simulations have been indispensable in assessing predictions of such theories, and invalidated the isotropic monoexponential character of the quadrupolar relaxation that is often assumed under a continuous-solvent description. A pronounced role of intermolecular cross-correlations on the relaxation was emphasized[17,29,38]. While classical MD often relies on the Sternheimer approximation to incorporate electron cloud effects[39,42–44], ab initio methods provide the best accuracy of the computed EFG at the ion position[30,32,45–47]. However, the associated high computational cost often impedes the long-time sampling of EFG fluctuations[30,31] and the accuracy of the correlation time estimates, even for aqueous ions at infinite dilution[30,32]. Hence, uncertainties arising both in ab initio and classical MD-based approaches hindered the quantitative comparison with experimental NMR relaxation rates and systematic analysis of the quadrupolar relaxation mechanisms.

Here we show that applying ab initio calculations to parametrize $C_Q$ in conjunction with classical MD to evaluate $\tau_c$ allows reaching a good agreement between the calculated and experimentally-obtained quadrupolar rates of $^{23}$Na$^+$ in electrolyte solutions over a broad range of salt concentrations and temperatures, thereby enabling a systematic analysis of the relaxation pathways and detailed modeling of the underlying dynamics. We find that the main effect of increased relaxivity is due to a lengthening of the correlation times, rather than a change of the average quadrupolar coupling constant. Counterintuitively, the latter varies mildly over the range of considered parameters, slightly decreasing with concentration and increasing with temperature. We conclude that, contrary to the commonly-assumed picture, rotational models based on the water dipole reorientation or Stokes-Einstein-Debye relation significantly overestimate the EFG correlation times. Rather, our results indicate that the EFG relaxation is mainly determined by the dynamics in the first two solvation shells around the solute and occurs over a time scale comparable to that of solution structural rearrangements. This work thus suggests that the subpicosecond collective dynamics of the liquid primarily drive the quadrupolar relaxation at the sodium-ion position, thereby offering insights into the quadrupolar relaxation mechanisms in electrolyte solutions.

## Results
### Electron cloud contribution to electric field gradients
We perform density functional theory (DFT) calculations to determine quantum EFGs at the Na$^+$ position in aqueous sodium chloride (NaCl) solutions at varying salt concentrations $c = 1$–5 molal (denoted with mol kg$^{-1}$ or m) at $T = 25\,^\circ$C (see Methods for details). The projector-augmented wave (PAW) method[45,47,48] is used to reconstruct the all-electron charge density at the nucleus. A configuration of a NaCl solution at 5 mol kg$^{-1}$ with converged charge densities is highlighted in Fig. 1a.

In classical MD, the electron cloud contribution to the EFG can be incorporated by means of the Sternheimer approximation[43,44], in which the full EFG at the nucleus $\mathbf{V}$ is proportional to the EFG created by the external charge distribution $\mathbf{V}_{\text{ext}}$: $\mathbf{V} \simeq (1 + \gamma)\mathbf{V}_{\text{ext}}$. Here, the electron cloud polarization is included via the simple rescaling factor $1 + \gamma$, with the Sternheimer (anti)shielding factor $\gamma$ being typically large $\gamma \gg 1$[44]. To validate the Sternheimer approximation for present systems, we have compared ab initio, $\mathbf{V}_{\text{AI}}$, against classical, $\mathbf{V}_{\text{ext}}$, EFGs at the Na$^+$ position, as determined on the same set of classically generated solution configurations (see Methods). Consistently with aqueous ions at infinite

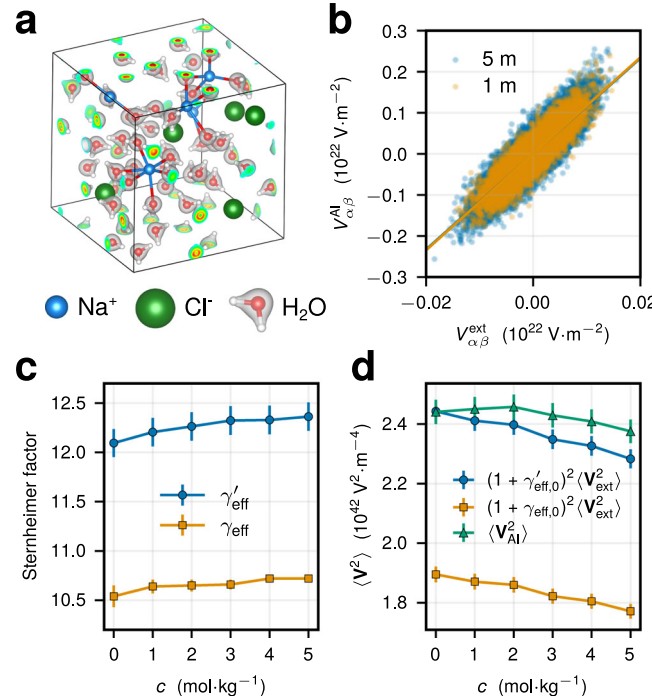

**Fig. 1 | Electron cloud contribution to the EFG at the Na$^+$ position.**
**a** Representative system configuration of a NaCl solution at 5 m. The gray opaque regions around water molecules show charge densities obtained with DFT PAW calculations (see Methods). **b** Component-wise comparison of ab initio EFGs, $V^{\text{AI}}_{\alpha\beta}$, against classical EFGs, $V^{\text{ext}}_{\alpha\beta}$, at the position of Na$^+$ ions on the same set of configurations for different salt concentrations $c$. The solid lines indicate the fit for an effective Sternheimer factor $\gamma_{\text{eff}}$: $V^{\text{AI}}_{\alpha\beta} = (1 + \gamma_{\text{eff}})V^{\text{ext}}_{\alpha\beta}$. **c** Effective Sternheimer factors for Na$^+$ obtained from the linear fit (yellow squares) or from the ratio $(1 + \gamma'_{\text{eff}})^2 = \langle \mathbf{V}^2_{\text{AI}} \rangle / \langle \mathbf{V}^2_{\text{ext}} \rangle$ (blue circles) at different $c$. **d** EFG variance at the Na$^+$ position for different $c$ as obtained directly with ab initio calculations (green triangles), or using the value of $\gamma_{\text{eff}}$ (yellow squares) and $\gamma'_{\text{eff}}$ (blue circles) at infinite dilution. The error bars in **c**, **d** were calculated using bootstrapping.

dilution[39,42], we find a strong correlation between $\mathbf{V}_{\text{AI}}$ and $\mathbf{V}_{\text{ext}}$ for all $c = 1$–5 mol kg$^{-1}$, as seen in Fig. 1b for the two extreme cases. The latter allows us to define effective Sternheimer factors $\gamma_{\text{eff}}$ through the linear fit $V^{\text{AI}}_{\alpha\beta} = (1 + \gamma_{\text{eff}})V^{\text{ext}}_{\alpha\beta}$. As seen in Fig. 1c, the resulting $\gamma_{\text{eff}}$ feature a small increase with $c$ (<5% compared to the infinite dilution value $\gamma_{\text{eff},0} = 10.54 \pm 0.11$[42]) associated with the modifications of the ion's solvation sphere (see Supplementary Note 5 and 6).

Despite the small changes of $\gamma_{\text{eff}}$ with increasing $c$, the Sternheimer approximation for the EFG variance, $(1 + \gamma_{\text{eff}})^2 \langle \mathbf{V}^2_{\text{ext}} \rangle$, that is necessary for the NMR relaxation rate computation (Eq. (1) in Methods) underestimates the ab initio value $\langle \mathbf{V}^2_{\text{AI}} \rangle$ by more than 20% (highlighted in Fig. 1d using $\gamma_{\text{eff},0}$). This again underlines the deficiencies of the Sternheimer approximation[42] that does not take into account non-electrostatic electron cloud polarization effects, such as short-range repulsion[18,49,50]. To improve upon the variance predictions, we formally define the Sternheimer factor $\gamma'_{\text{eff}}$ as $(1 + \gamma'_{\text{eff}})^2 = \langle \mathbf{V}^2_{\text{AI}} \rangle / \langle \mathbf{V}^2_{\text{ext}} \rangle$ with state-dependent values of $\langle \mathbf{V}^2_{\text{AI}} \rangle$ and $\langle \mathbf{V}^2_{\text{ext}} \rangle$. Similarly, $\gamma'_{\text{eff}}$ slowly grows with $c$, yet starting from a markedly enhanced value of $\gamma'_{\text{eff},0} = 12.09 \pm 0.14$ at infinite dilution (Fig. 1c). The EFG variance prediction $(1 + \gamma'_{\text{eff},0})^2 \langle \mathbf{V}^2_{\text{ext}} \rangle$ using $\gamma'_{\text{eff},0}$ at infinite dilution is within 5% accuracy of $\langle \mathbf{V}^2_{\text{AI}} \rangle$ within the considered concentration range, a much better estimate in comparison to the simple Sternheimer approximation (Fig. 1d). While not capturing all condensed-phase effects that arise with increasing $c$, the estimate $(1 + \gamma'_{\text{eff},0})^2 \langle \mathbf{V}^2_{\text{ext}} \rangle$ provides a fair accuracy, reproduces the trend of $\langle \mathbf{V}^2_{\text{AI}} \rangle$ to decrease with the salt concentration (see Fig. 1d), and permits avoiding computationally expensive DFT calculations at multiple system state points of interest.

As discussed below, in combination with the EFG relaxation dynamics captured at the classical level, this approach provides a good description of the quadrupolar $^{23}$Na$^+$ NMR relaxation rates in aqueous solutions.

### Relaxation of electric field gradient fluctuations

We perform classical MD simulations employing the Madrid-2019 force field (FF)[51,52] to facilitate the long-time sampling of EFG fluctuations and to investigate the mechanisms behind the concentration and temperature behavior of the quadrupolar $^{23}$Na$^+$ NMR relaxation rate in aqueous sodium chloride, bromide (NaBr), and fluoride (NaF) solutions (see Methods for simulation details). Two facts give confidence in this approach: (i) a very strong correlation between the full and classical (external) EFGs (Fig. 1b), indicating that the dynamics of the former should be largely determined by that of the latter; (ii) while classical MD with rigid water molecules do not quantitatively reproduce the librational or hydrogen-bond stretching water dynamics that occur at very short times below ~ 50 fs[53], it is expected that these high-frequency motions do not significantly affect the dominating long-time (~1 ps) EFG relaxation mode (e.g., see ref. 40 and below).

We provide a systematic description of the $^{23}$Na$^+$ EFG relaxation at short, intermediate, and long times ranging from a few fs to tens of ps. Increasing salt concentration $c$ or decreasing temperature $T$ causes a profound slow-down of the EFG fluctuations at the ion position (Fig. 2). Due to a qualitative similarity of the EFG relaxation in the solutions considered, here we will focus on the case of NaCl; see Supplementary Information (SI) for NaBr and NaF. Figure 2a shows the autocorrelation functions (ACFs) of the classical EFG at the Na$^+$ position, $C_{EFG}(t) \equiv \langle \mathbf{V}_{ext}(0) : \mathbf{V}_{ext}(t) \rangle$, as a function of $c$ at $T = 25\,°C$ (see Supplementary Fig. 9 for other $T$). Similarly to a single Na$^+$ in water[32,38,39,42], $C_{EFG}(t)$ relaxes in two steps: (i) a rapid initial decay happening at $t \lesssim 0.2$ ps that corresponds to ≈70% of the EFG decorrelation. This is in good agreement with the ab initio MD results for Na$^+$ at infinite dilution[32], highlighting the validity of the classical approach; (ii) a much slower secondary decay occurring in the picosecond regime. As seen in Fig. 2a, the increase in $c$ leaves the initial fast decay practically unchanged, while causing a pronounced slow-down of the second decay mode. The latter is highlighted in the inset of Fig. 2a showing the EFG ACFs for $t < 1$ ps for different $c$ at $T = 25\,°C$ (see also Supplementary Fig. 10). A qualitatively similar trend is found with decreasing temperature, as we show in Fig. 2b at $c = 4$ m and in Supplementary Fig. 9 for other $c$.

The form of the EFG ACF decay in Fig. 2 suggests a collective pathway behind the relaxation. After the initial fast decay that can be described with an exponential $\sim e^{-t/\tau_f}$ with $\tau_f \approx 62$ fs, we find a development of a much slower relaxation mode that profoundly depends on $c$ and $T$. Compared to earlier results[40], our long-time sampling reveals that the slow part of the EFG ACF is not exponential, as clearly seen from the behavior of $C_{EFG}(t)$ on a semi-logarithmic scale in Fig. 2a–b and as we show with explicit fits in Supplementary Note 9. Except at very long times, we find that the slow decay can be modeled either with a two-exponential (Supplementary Fig. 11) or a stretched exponential fit $\sim e^{-(t/\tau_s)^\beta}$ with $\beta = 0.67 \pm 0.05$ (Supplementary Figs. 12 and 13), which suggests a broad distribution of contributing relaxation modes (Supplementary Fig. 14). Although observed over a limited time range (up to a decade), we find that the long-time tail of the EFG ACFs is consistent with a power-law $\sim t^{-5/2}$, as shown with $C_{EFG}(t)$ on a log-log scale for $c = 4$ m in Fig. 2c and with $t^{5/2}C_{EFG}(t)$ in Supplementary Fig. 15. Such a hydrodynamic tail was predicted by a mode-coupling theory of Bosse et al. for the EFG ACF in molten salts[25]. It originates from the coupling between the ion motion and shear excitations in the liquid, a mechanism causing the well-known $\sim t^{-3/2}$ tail of the velocity ACF[54]. While sampling of the EFG fluctuations at even longer time scales is necessary to decisively confirm to presence of $\sim t^{-5/2}$ regime, our results for Na$^+$ in Fig. 2c suggest that its relative contribution may be marginal

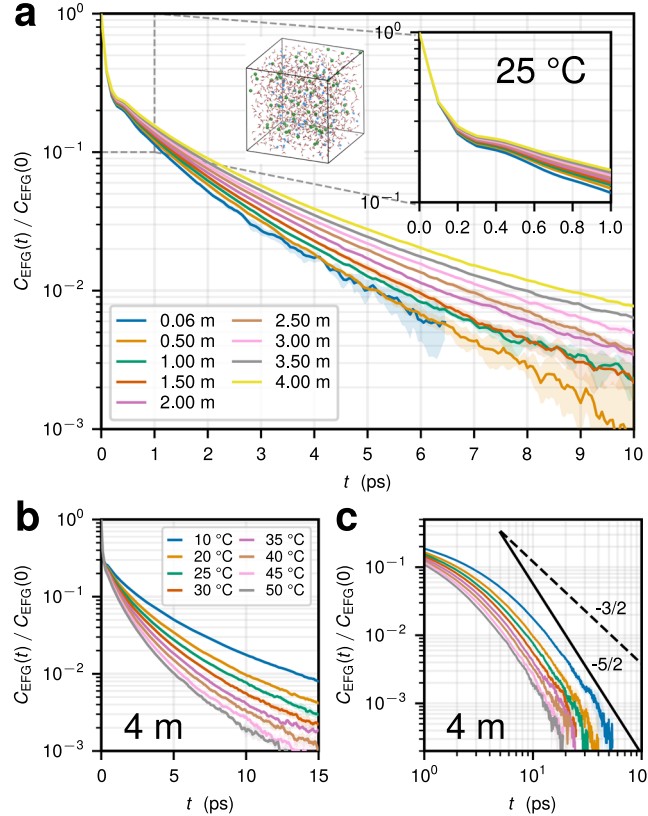

**Fig. 2 | Relaxation of EFG fluctuations. a** Normalized autocorrelation functions $C_{EFG}(t)/C_{EFG}(0)$ of the EFG at the position of a Na$^+$ ion obtained using classical MD simulations for different salt concentrations $c$ at $T = 25\,°C$ in aqueous NaCl solutions ($c$ increases from bottom to top). Qualitatively similar trends are found for other concentrations and temperatures (see Supplementary Fig. 9). Insets in **a** highlight the short-time behavior of the ACFs for $t < 1$ ps and a typical system configuration at $c = 4$ m (Na$^+$ and Cl$^-$ ions are blue and green, respectively). **b** Temperature behavior of $C_{EFG}(t)$ at $c = 4$ m ($T$ decreases from bottom to top). **c** Long-time behavior of $C_{EFG}(t)$ plotted on a double logarithmic scale at $c = 4$ m for different temperatures (the legend shown in **b**). The black solid and dashed lines highlight a power-law scaling $\sim t^\alpha$ with $\alpha = -5/2$ and $\alpha = -3/2$, respectively. See also Supplementary Fig. 15 for $C_{EFG}(t)$ multiplied by $t^{5/2}$ and $t^{3/2}$. Shaded regions in **a**–**c** indicate standard errors from multiple independent simulation runs.

because the apparent onset of the algebraic decay occurs at times when the ACF has decayed considerably.

### Quadrupolar relaxation rates

The combination of EFG fluctuations captured at the classical level and consistent inclusion of the electron cloud contribution to the EFG enables reaching a good quantitative agreement between the calculated and experimentally-measured quadrupolar NMR relaxation rate for $^{23}$Na$^+$ in aqueous NaCl, as we compare in Fig. 3 with filled and open symbols, respectively. As seen in Eq. (1) in Methods, the quadrupolar relaxation rate is proportional to the product of the effective correlation time of EFG fluctuations, $\tau_c = C_{EFG}^{-1}(0) \int_0^\infty dt\, C_{EFG}(t)$, and the EFG variance, which we approximate as $\langle \mathbf{V}^2 \rangle = (1 + \gamma'_{eff,0})^2 \langle \mathbf{V}_{ext}^2 \rangle$ with $\gamma'_{eff,0} = 12.09$ and $\langle \mathbf{V}_{ext}^2 \rangle = C_{EFG}(0)$. The integration of $C_{EFG}(t)$ over tens of picoseconds is necessary to obtain well-converged correlation times $\tau_c$ (Supplementary Fig. 16), notably at high salt concentrations and low temperatures (Fig. 2). Finally, our estimates in Supplementary Note 3 for the dipole-dipole contribution to the $^{23}$Na$^+$ rate $1/T_1$ due to interactions with the spins of $^1$H, $^{23}$Na, and $^{35}$Cl are more than four orders of magnitude smaller compared to the quadrupolar contribution, indicating that the latter dominates the $^{23}$Na NMR relaxation.

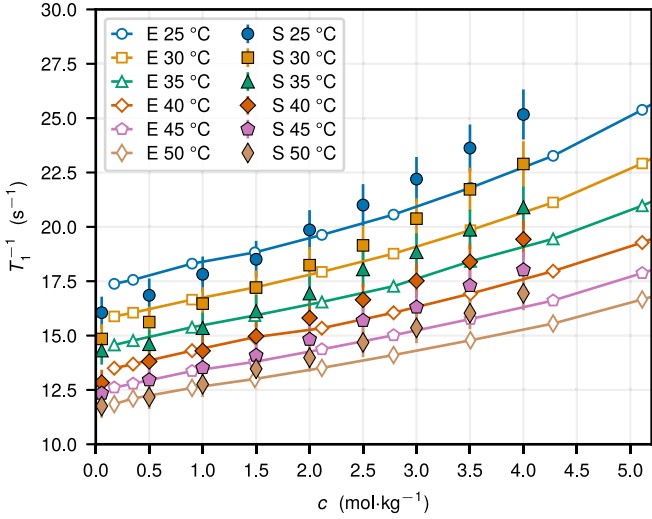

**Fig. 3 | Concentration and temperature dependence of the quadrupolar relaxation rate.** $T_1^{-1}$ of $^{23}Na^+$ in aqueous NaCl as a function of the salt concentration $c$ as obtained in experiments (E, solid lines and open symbols) and simulations (S, filled symbols) for different temperatures. The error bars for simulation results are associated with the approximation for incorporating the electron cloud contribution to the EFG.

The NMR relaxation rate $1/T_1$ grows with increasing salt concentration $c$ and with reducing temperature $T$ (Fig. 3). Under the extreme narrowing condition, which is fulfilled for the considered cases (see Methods), $1/T_1 \propto \tau_c$, thereby suggesting that the slowing down of EFG fluctuations (Fig. 2), as reflected in the augmented correlation time, determines the rate behavior. Experimentally, $1/T_1$ rises by about 50% within the considered range of concentrations $c = 0.17-5.1$ m for temperatures $T = 20-50$ °C, in line with the previous results[22,28,55,56]. At $T = 30$ °C, $1/T_1$ increases from around $15.9$ s$^{-1}$ at $c = 0.17$ m to $25.2$ s$^{-1}$ at $c = 5.1$ m. With increasing $T$ from 25 °C to 50 °C, $1/T_1$ reduces by >25% for considered salt concentrations. In general, our computational results for $1/T_1$ of $^{23}Na^+$ agree well with the experimental data, especially at lower salt concentrations $c \lesssim 2.5$ m, reproducing both the concentration and temperature behavior. For higher salt concentrations, we find that $1/T_1$ grows systematically faster with increasing $c$ as compared to the experiments, yet the relative error remains less than 15% over the considered range of conditions. The latter difference is likely caused by the shortcomings of the employed FF in capturing dynamic properties of the solution for $c \gtrsim 2$ m[52].

**Microscopic parameters of the relaxation**
We find that the slowing down of EFG fluctuations at the Na$^+$ position primarily causes a marked increase in the quadrupolar NMR relaxation rate with increasing $c$ and decreasing $T$ (Fig. 3). In Fig. 4, we quantify the role of dynamic and static effects that are reflected in the changes of $\tau_c$ and $\langle V^2 \rangle$, respectively, with varying salt concentration and temperature, as obtained in MD simulations of aqueous NaCl (see Supplementary Fig. 20 for other electrolyte solutions). While $\tau_c$ increases by a factor of ~1.5–2.5 with increasing $c$ and decreasing $T$ within the considered range of parameters (Fig. 4a), the value of $\langle V^2 \rangle$ reduces concurrently by up to 10% (Fig. 4c), indicating that the augmented correlation times are mainly responsible for the rate behavior.

For considered $c$ and $T$, $\tau_c$ of Na$^+$ is quite short and below 1 ps (Fig. 4a), a feature already pointed out in previous classical[35,38,39,42] and ab initio[30,32] MD studies at infinite dilution. At $T = 25$ °C, we find that $\tau_c$ increases from 0.41 ps at $c \approx 0.06$ m to 0.65 ps at $c = 4$ m. Despite the rapid decorrelation of EFG ACFs for $t \lesssim 0.2$ ps (Fig. 2), we find that the contribution of the slow relaxation process to $\tau_c$ yields >85% of its overall value and also grows with increasing $c$ and decreasing

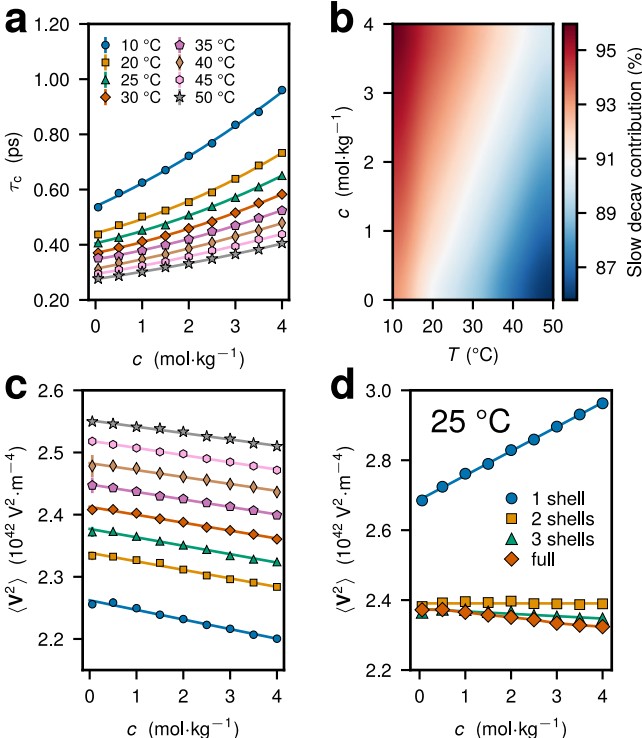

**Fig. 4 | Microscopic parameters of the EFG relaxation. a** Effective correlation time $\tau_c$ of the EFG fluctuations at the Na$^+$ position as a function of salt concentration $c$ for different temperatures $T$. **b** Relative contribution of the slow EFG relaxation mode to $\tau_c$ for different $c$ and $T$. The contribution was estimated using the stretched exponential fit of the normalized EFG ACFs (Supplementary Note 9). **c** Variance of the total EFG at the ion position $\langle V^2 \rangle = (1 + \gamma'_{eff,0})^2 \langle V_{ext}^2 \rangle$ as a function of $c$ for different temperatures. The legend in **c** is the same as in **a**. **d** EFG variance, $\langle V^2 \rangle$, evaluated from water molecules and ions located within a different number of solvation shells around the central Na$^+$ ion as a function of $c$ at $T = 25$ °C. The standard error from multiple independent simulation runs is either explicitly shown or does not exceed the symbol size.

$T$ (Fig. 4b). The dominance of the slow non-exponential decay of EFG ACFs over the $\tau_c$ behavior again exemplifies the governing role of collective processes behind the quadrupolar Na$^+$ relaxation.

While the EFG variance at the Na$^+$ position is largely determined by the first solvation shell contribution (Fig. 4d), a quantitative understanding of the QCC is only achieved if we take into account point charges within a radius of $r \gtrsim 8$ Å around the central ion, approximately the length scale of pronounced ion-ion and ion-solvent correlations (Supplementary Figs. 3 and 4). Similarly, we find that the EFG relaxation dynamics is well captured by point charge contributions located within the first two solvation shells around the ion, whereas the EFG due to the first solvation shell relaxes much more slowly (Supplementary Fig. 8). The first two solvation shells of Na$^+$ are predominantly populated by water molecules even at the highest $c = 4$ m considered (see Supplementary Note 5), suggesting that the solvent provides the largest contribution to the EFG at the Na$^+$ position and that other ions mostly retard water dynamics.

We observe that $\langle V^2 \rangle$ is reduced in bipyramidal complexes with octahedral symmetry, coordinated by six water molecules, yet only by 10% compared to the ensemble average (Supplementary Note 7). The contribution of the first solvation shell to the EFG variance features an increase with $c$ (Fig. 4d), correlated with the fact that the six-coordinated state becomes less likely with increasing the salt concentration (Supplementary Fig. 7). Our consistently calculated QCC for $^{23}Na^+$ in aqueous NaCl is in the range between $19 \times 10^6$ and $20.6 \times 10^6$ rad × s$^{-1}$ for considered $c$ and $T$ (Supplementary Fig. 17), a

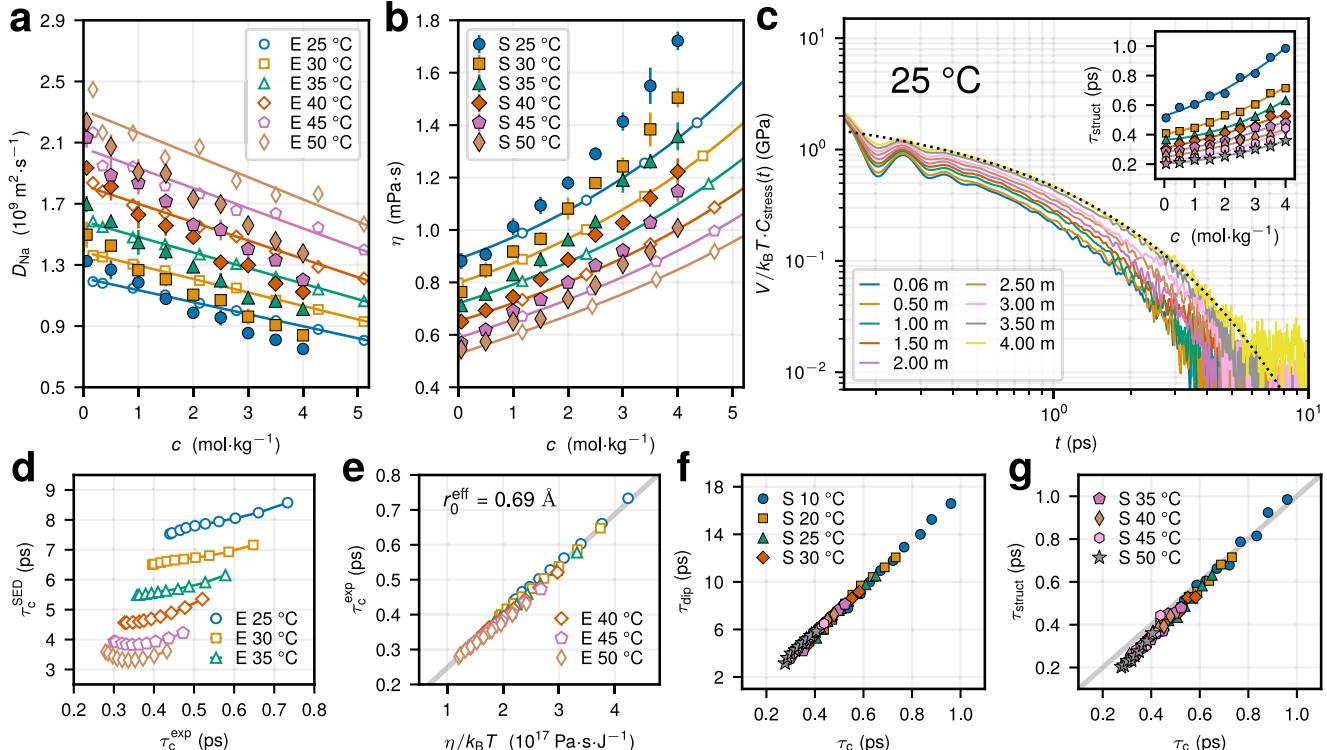

**Fig. 5 | Assessing models of the quadrupolar relaxation. a** Na$^+$ diffusion coefficient as a function of salt concentration $c$ for different temperatures $T$ in experiments (open symbols, legend shown in **a**) and simulations (filled symbols, legend shown in **b**). **b** Dynamic solution viscosity as a function of $c$ for different $T$ in experiments (open symbols, legend shown in **a**) and simulations (filled symbols, legend shown in **b**). Experimental viscosities were taken from ref. 60. Solid lines in **a**, **b** are polynomial fits of the experimental data. **c** Stress tensor ACFs $C_{stress}(t)$ normalized by the system volume $V$ and $k_B T$ for increasing $C$ at $T = 25$ °C. The black dotted line shows the stretched exponential fit of the long-time decay at $c = 4$ m. Inset: the time scale of solution structural relaxation $\tau_{struct}$, as extracted from the long-time decay of $C_{stress}(t)$ (see main text), as a function of $c$ at multiple temperatures. $T$ decreases from top to bottom, and the legend is shown in **f**, **g**. **d** Stokes-Einstein-Debye time plotted versus the EFG correlation time $\tau_c^{exp}$, as extracted from experimental data for different $c$ and $T$. **e** $\tau_c^{exp}$ as a function of $\eta/k_B T$ for different temperatures. $r_0^{eff}$ is the effective hydrodynamic radius of a Na$^+$ ion extracted under assumption that $\tau_c^{exp}$ can be modeled by a SED relation (the gray line shows the best fit). **d**, **e** Share the same legend. **f** Mean water dipole reorientation time $\tau_{dip}$ plotted versus the EFG correlation time $\tau_c$, as extracted in simulations for different $c$ and $T$. **g** $\tau_{struct}$ plotted versus $\tau_c$ for different $c$ and $T$. The gray line indicates the linear dependence, $\tau_{struct} = \tau_c$. **a–c**, **f**, **g** The standard error from independent simulation runs are either explicitly shown or do not exceed the symbol size.

value ~3–4 times larger than previous estimates based on the assumption that the EFG primarily decorrelates by translational and reorientational water dynamics with $\tau_c \approx 3 - 7$ ps[19–22,57]. We thus conclude that the aforementioned modes of motion provide only a minor contribution to the observed relaxation.

## Assessment of the relaxation models

We utilize the information available in experiments and molecular simulations in Fig. 5 to shed light on the mechanisms behind the quadrupolar relaxation. First, we focus on the possibility to model the EFG correlation time $\tau_c$ using the commonly used Stokes–Einstein–Debye (SED) relation $\tau_c^{SED} = 4\pi\eta r_0^3/3k_B T$, where $\eta$ is the dynamic viscosity of the solution, $r_0$ is the sodium's hydrodynamic (Stokes) radius, and $k_B$ is the Boltzmann constant. Within the SED picture, the EFG relaxation at the Na$^+$ position is governed by the Brownian rotational diffusion, likely to be related with collective reorientations of ion-water solvation complexes[55]. While the SED model assumptions are not expected to hold down to the molecular scale[58,59], we systematically explore $\tau_c^{SED}$ in relation to $\tau_c$, as it is often exploited to rationalize quadrupolar relaxation dynamics of $^{23}$Na$^+$ [19–21].

We use the translational Stokes-Einstein relation $D = k_B T/6\pi\eta r_0$ to determine the concentration- and temperature-dependent values of the Stokes radius from the experimental Na$^+$ diffusion coefficients (Fig. 5a) and highly accurate NaCl viscosity values provided by Kestin et al.[60] (Fig. 5b). $D$ and $\eta$ calculated in our MD simulations (see Methods) are in good agreement with the experiments, especially

for $c \lesssim 2$ m, capturing both the concentration and temperature behavior (compare filled and open symbols in Fig. 5a, b). The viscosity $\eta$ in MD was obtained via the Green-Kubo formula using the stress ACFs (Fig. 5c), as detailed in Eqs. (3) and (4) in Methods. In Fig. 5d, we compare $\tau_c^{SED}$ calculated from state-dependent Stokes radii $r_0(c, T)$ against the effective EFG correlation time $\tau_c^{exp}$ obtained from the experimental NMR relaxation rates and the sodium's QCC from simulations (Supplementary Fig. 17) rather than those from previous estimates[19–22,57]. $r_0$ assumes values between 1.5 and 2.0 Å for considered parameters (Supplementary Fig. 19). While both $\tau_c^{SED}$ and $\tau_c^{exp}$ generally lengthen with increasing $c$ and decreasing $T$, $\tau_c^{SED}$ exceeds $\tau_c^{exp}$ by a factor of 8–17 (Fig. 5d). Similar results are obtained in our simulations (Supplementary Fig. 19). At $T = 25$ °C, $\tau_c^{SED}$ increases from around 7.5 to 8.2 ps, larger by more than one order of magnitude than $\tau_c^{exp}$ that grows from 0.44 to 0.66 ps for increasing $c$ from 0.17 to 5.1 m. Thus, we conclude that the EFG correlation times cannot be understood on the basis of the SED relation parameterized using the translational hydrodynamic radius of sodium ions $r_0 = k_B T/6\pi\eta D$.

This is further illustrated in Fig. 5e showing $\tau_c^{exp}$ plotted against $\eta/k_B T$ for various temperatures. While a Stokes-Einstein-like relation holds for $\tau_c^{exp}$, that is a strong correlation $\tau_c^{exp} \propto \eta/k_B T$ exists for the considered range of parameters, the effective Stokes radius $r_0^{eff} = 0.69$ Å that would correspond to the EFG correlation time $\tau_c^{exp}$ within the SED model is clearly unphysical and smaller than the ionic radius 1.02 Å. $r_0^{eff}$ was obtained from the fit $\tau_c^{exp} = 4\pi\eta [r_0^{eff}]^3/3k_B T + \tau_0^{eff}$ with an additional intercept $\tau_0^{eff} = 0.11$ ps needed for the best data representation[59].

We obtain a similar value of $r_0^{eff} \approx 0.68$ Å from our MD simulations (Supplementary Fig. 19). Therefore, the validity of the relation $\tau_c^{exp} \propto \eta / k_B T$ explains the correlation between $\tau_c^{exp}$ and $D^{-1}$ reported in refs. [19–21], rather than simplified assumptions of the rotational Brownian diffusion that yield much larger estimates of $\tau_c^{exp}$ (Fig. 5d).

We now return to microscopic time scales of molecular motion in relation to that EFG fluctuations. The average water dipole reorientation time $\tau_{dip} = \int_0^\infty dt \langle P_1[\mathbf{u}(t) \cdot \mathbf{u}(0)] \rangle$ assumed to drive the quadrupolar relaxation within the Hertz model[17,26,27,38] is 11-14 times larger compared to $\tau_c$, as extracted in our simulations (Fig. 5f and Supplementary Fig. 18). Above, $\mathbf{u}$ is a unit vector pointing along the HOH bisector of a water molecule and $P_1(x) = x$ is the first Legendre polynomial. This indicates that the single molecule reorientation with neglected intermolecule cross-correlations cannot explain the EFG relaxation dynamics. Yet, as seen in Fig. 5f, both $\tau_{dip}$ and $\tau_c$ increase similarly with increasing $c$ and decreasing $T$, suggesting that the overall deceleration of the electrolyte dynamics, marked by an enhanced viscosity, impacts in a similar way both the motions that drive water reorientation as well as those that cause the EFG relaxation at the ion position.

To illustrate the relationship between these effects, in our MD simulations we extract a typical time scale of solution structural relaxation $\tau_{struct}$ using the stress tensor ACFs (Fig. 5c). While the short-time behavior of $C_{stress}(t)$ corresponding to elastic, vibrational contributions features little changes with varying $c$ and $T$[61], its long-time tail slows down with increasing $c$ and decreasing $T$, indicating an overall deceleration of the viscous dynamics of the liquid. We find that the long-time tail can be modeled well using a stretched exponential decay, $\sim e^{-(t/\tau_K)^{\beta_K}}$, with $\beta_K \approx 0.61 \pm 0.04$ (consistent with earlier simulations of pure water[61] and time-resolved spectroscopy experiments[62]). $\tau_K$ is in the range between 0.13 and 0.70 ps for considered parameters. The mean structural relaxation time $\tau_{struct} = \tau_K \beta_K^{-1} \Gamma(\beta_K^{-1})$, defined through the integral of the stretched exponential expression, is strongly correlated and comparable to the subpicosecond EFG correlation time $\tau_c$ (Fig. 5g). While the stress and EFG tensors are not directly related to each other, both quantities are inherently collective, that is the relaxation of their fluctuations is mainly driven by many-body correlations, and features a similar stretched decay for $t \gtrsim 0.4$ ps. All these observations suggest that the fast collective dynamics of the liquid that drive its structural rearrangements are also responsible for the quadrupolar NMR relaxation.

## Discussion

We have shown that the multiscale methodology combining DFT PAW calculations to parameterize the QCC and classical MD simulations to sample long-time EFG fluctuations enables an accurate description of the quadrupolar NMR relaxation rates of $^{23}$Na$^+$ in aqueous electrolyte solutions over a broad range of salt concentrations and temperatures. The resulting NMR relaxation rates are in very good agreement with the experimental data, especially at low salt concentrations, as validated in aqueous NaCl at multiple system state points. We find that the growth of the relaxation rate $T_1^{-1}$ with increasing $c$ and decreasing $T$ is primarily due to the slowing down of the EFG fluctuations reflected in the augmented EFG correlation time $\tau_c$, while the concurrent changes in the QCC are rather small. The availability of dynamic information over a broad range of system parameters enabled us to have a consistent discussion concerning the quadrupolar relaxation models. We have demonstrated that the commonly-assumed rotational relaxation models based on either the water dipole reorientation[26–28] or on the Stokes-Einstein-Debye relation[19–21] overestimate the consistently-determined $\tau_c$ by at least an order of magnitude. This disagreement is understandable as these models restrict the relaxation description to one- or two-body correlations, oversimplifying the collective dynamics of the intermolecular EFG at the ion position[17,29,38]. The quantitative interpretation of the EFG correlation times in terms of such simple isotropic models should therefore be used with caution. Instead, our results indicate that the Na$^+$ EFG relaxation is largely determined by the dynamics in the two first solvation shells of the ion and occurs over a subpicosecond time scale comparable to that of solution structural rearrangements $\tau_{struct}$, as extracted from the relaxation of the stress tensor. This again invalidates a continuous-solvent hydrodynamic description assuming that $\tau_c \gg \tau_{struct}$.

Rather than directly probing single ion diffusion or single water molecule reorientation, our results suggest that the quadrupolar NMR relaxometry of $^{23}$Na$^+$ may be used as a complementary tool to analyze electrolyte dynamics in the THz domain, as relevant for emerging sodium-ion battery technologies[9–11]. As quadrupolar relaxation is largely determined by the processes in the immediate vicinity of the solute, it can provide supplementary information on the fast, collective, molecular motions in ionic solvation cages that have been associated with the high-frequency dielectric response[63], solvation dynamics[64], as well as structural relaxation[61] in aqueous electrolytes. The ability to capture the NMR relaxation rates by means of classical MD allows elucidating the quadrupolar relaxation mechanisms that occur in multicomponent systems, such as concentrated aqueous solutions of multiple salts[65], mixtures of salts with glycerol[66], or polyelectrolytes[67], where the relaxation dynamics may be influenced by environment heterogeneity, interface formation, microphase separation, or ion binding to polyelectrolyte chains. Future work could also focus on developing mesoscopic approaches that would allow a quantitative description of quadrupolar relaxation in complex biological-type compartments, characterized either by slow-motion conditions with dynamics within intracellular and extracellular spaces in biological tissues that may include structural anisotropy with residual quadrupolar coupling and distribution of correlation times (as, for example, in connective tissue where sodium ions are surrounded by a collagen matrix[4]). Such models would further provide a foundation for the interpretation of magnetic resonance imaging contrast mechanisms that are sensitive to the quadrupolar interaction[68–71].

## Methods

### Quadrupolar NMR relaxation rates

The quadrupolar mechanism dominates the relaxation of nuclei with spin $I > 1/2$ and is due to the coupling between their quadrupolar moment $eQ$ with the EFG tensor $\mathbf{V}$ at the nucleus position[16]. While the NMR relaxation of spin components is generally bi-exponential for $^{23}$Na with $I = 3/2$[22,72], it is possible to define effective longitudinal and transverse quadrupolar relaxation rates, $1/T_1$ and $1/T_2$, respectively, provided that the "fast motion" or "extreme narrowing" regime holds[30,73,74]. In this case, the characteristic EFG correlation time $\tau_c$ is much smaller than the Larmor period $\omega_0^{-1}$ of the nucleus, $\omega_0 \tau_c \ll 1$. The latter can be shown to be fulfilled for all cases considered in this work as the relevant correlation times for $^{23}$Na in electrolyte solutions are below 100 ps, and the magnetic field used in the experiments is 11.7 T that corresponds to $\omega_0^{-1} \approx 7.6$ ns. As we show in more detail in the SI, the two quadrupolar relaxation rates become equal in the fast motion regime and, combined with the rotational invariance of the system, can be expressed as[38]

$$\frac{1}{T_1} = \frac{2I + 3}{20 I^2 (2I - 1)} \left( \frac{eQ}{\hbar} \right)^2 \langle \mathbf{V}^2 \rangle \tau_c \tag{1}$$

where $\hbar$ is the reduced Planck constant, $\tau_c$ is an effective correlation time of EFG fluctuations

$$\tau_c = \langle \mathbf{V}^2 \rangle^{-1} \int_0^\infty dt \langle \mathbf{V}(0) : \mathbf{V}(t) \rangle, \qquad (2)$$

where $\langle \mathbf{V}(0) : \mathbf{V}(t) \rangle = \sum_{\alpha,\beta} \langle V_{\alpha\beta}(0) V_{\alpha\beta}(t) \rangle$ with $\alpha$, $\beta = x$, $y$, $z$ and the brackets $\langle \ldots \rangle$ denoting an ensemble average, and $\langle \mathbf{V}^2 \rangle \equiv \langle \mathbf{V}(0) : \mathbf{V}(0) \rangle$ is the EFG variance at the ion position. For a $^{23}$Na nucleus with $I = 3/2$ and $Q = 104 \times 10^{-31}$ m$^2$ [75], the rate constant $1/T_1$ can be recast as $1/T_1 = C_Q^2 \tau_c / 10$ with the quadrupolar coupling constant (QCC) defined as $C_Q^2 = \frac{2}{3} \left( \frac{eQ}{\hbar} \right)^2 \langle \mathbf{V}^2 \rangle$ [16,30]. Finally, Eq. (1), which follows from linear response theory, allows calculating the quadrupolar spin-lattice relaxation rate $1/T_1$ from the EFG fluctuations in equilibrium MD simulations without an imposed magnetic field.

## NMR experiments

Solution samples with nine different NaCl concentrations were prepared by mixing $x$ mg of NaCl in ($y$–$x$) mg of deionized water in a beaker, with $x = 0.1$, 0.2, 0.5, 0.8, 1.1, 1.4, 1.7, 2.0, 2.3 mg and $y = 10$ mg, to make solutions of concentrations 0.173, 0.349, 0.901, 1.488, 2.115, 2.786, 3.505, 4.278, and 5.111 mol kg$^{-1}$ to 5 mm NMR tubes (sample volume = 0.5 mL). All mass measurements were performed on a Mettler Toledo ME204E balance with a resolution of 0.1 mg. The solution at 26% weight corresponds to NaCl saturation in water at 20 °C [76].

NMR experiments were performed on an 11.7 T NMR Bruker Avance I spectrometer operating at 132.3 MHz for $^{23}$Na, using a 5 mm double resonance broadband probe. The test tubes with the solutions were placed inside the spectrometer where the sample temperature could be controlled using gas flow and a temperature sensor providing precise, stable, and reliable temperature regulation. After each desired temperature reached steady state, a standard free induction decay was acquired followed by a longitudinal relaxation time $T_1$ mapping sequence, and a diffusion pulse sequence. At each temperature, the tuning and matching was checked. The duration of the 90 ° pulse was 9.6 μs, whereas that for the 180 ° pulse was 19.6 μs. A standard inversion-recovery pulse sequence was used to acquire $T_1$ with 32 logarithmically spaced steps. The delay was varied from 1 ms to 400 ms for $^{23}$Na. Diffusion coefficients were measured using a Pulsed-Gradient-Spin-Echo in 32 steps with a maximum $b$-value of 2200 s mm$^{-2}$. The maximum diffusion gradient was 1 T m$^{-1}$ and the duration was 4 ms.

## Molecular dynamics simulations

Aqueous sodium chloride (NaCl), bromide (NaBr), and fluoride (NaF) solutions were simulated using classical MD employing the Madrid-2019 FF [51] that is based on the TIP4P/2005 water model [77] and uses scaled charges of $+0.85e$ and $−0.85e$ ($e$ is the fundamental unit of charge) for Na$^+$ cations and Cl$^-$, Br$^-$, and F$^-$ anions, respectively. The FF parameters are listed in the SI. The scaled ionic charges aim at taking into account the electronic contribution to the dielectric constant at high frequencies in a mean-field fashion [78]. At a moderate computational cost in comparison to fully polarizable models, the EFG relaxation within the Madrid-2019 FF [51] has recently been shown to accurately describe the quadrupolar NMR relaxation rates of alkali metal ions at infinite dilution [42], in particular, that of Na$^+$. Solutions comprised of $N = 1000$ water molecules and $N_p$ ion pairs were initialized at different salt concentrations $c$ between 0.06 m ($N_p = 1$) and 4 m ($N_p = 72$) in a cubic box at the equilibrium solution density $\rho(c, T)$ obtained in NPT simulations at $P = 1$ bar. The densities are in excellent agreement with the experimental ones, as discussed in Supplementary Note 2.

The equilibrated electrolyte systems were then simulated in the NVT ensemble. Both NPT and NVT simulation runs were carried out in

the open-source MetalWalls package on graphics processing units [79] with electrostatic interactions computed with Ewald summation [80] and a short-range cutoff of 1.24 nm. The constant temperature was maintained using the Nose-Hoover chains thermostat with a time constant of 1 ps. System temperatures in range from 10 °C to 50 °C were considered. The equations of motion were integrated using the velocity Verlet algorithm and an integration time step of 1 fs. The effective rigidity of water molecules was imposed with the help of the RATTLE algorithm with a precision of 10$^{-9}$. For each ($c$, $T$) state point, at least five independent runs of length 5 ns were performed to measure the EFG at the ion positions (sampled every 50 fs). Full Ewald summation expressions [80] were used in the computation of the EFGs, as recently implemented in MetalWalls [42]. For the considered system parameters, the relaxation of EFG fluctuations was found not to be affected by the finite box size, as we discuss in Supplementary Note 4.

## Ab initio calculations

To determine EFGs with the electron cloud contribution, smaller systems containing 55 water molecules and $N_p = 1, 2, 3, 4,$ and 5 NaCl ions pairs, corresponding to the salt concentrations $c = 1, 2, 3, 4, 5$ mol kg$^{-1}$, were simulated in the same way as the larger ones using the Madrid-2019 FF. In a single NVT simulation run at $T = 25$ °C, 2000 configurations were sampled with a period of 10 ps, and were later used in DFT-based EFG calculations with periodic boundary conditions in the Quantum Espresso (QE) package [81]. No additional geometry optimization of the configurations was performed in the DFT calculations. The pseudopotential-based projector-augmented wave (PAW) method [45,47,48] was used to reconstruct the all-electron charge density in the vicinity of the nucleus using the QE-GIPAW package [82]. The self-consistent electron densities were calculated using the PBE functional [83], a kinetic energy cutoff of 80 Ry, and norm-conserving pseudopotentials of the GIPAW package [84]. In the case of Na$^+$ ions, the EFGs obtained with the PBE functional were shown to be in good agreement [32] with those obtained with the hybrid PBE0 functional [85].

## Dynamical properties of electrolyte solutions

The shear viscosity of aqueous electrolyte solutions was obtained using the Green-Kubo relation [86]:

$$\eta = \frac{V}{k_B T} \int_0^{+\infty} dt \, C_{\text{stress}}(t), \qquad (3)$$

with $V$ being the system volume and $k_B$ standing for the Boltzmann constant. The stress tensor ACF $C_{\text{stress}}(t)$ was computed as [86]

$$C_{\text{stress}}(t) = \frac{1}{10} \sum_{\alpha,\beta} \langle P_{\alpha\beta}(t) P_{\alpha\beta}(0) \rangle, \qquad (4)$$

where $\alpha$, $\beta$ run over the three Cartesian components and $P_{\alpha\beta}$ is the traceless symmetrized part of the stress tensor $\sigma_{\alpha\beta}$: $P_{\alpha\beta} = \frac{1}{2} \left( \sigma_{\alpha\beta} + \sigma_{\beta\alpha} \right) - \frac{1}{3} \delta_{\alpha\beta} \sum_\gamma \sigma_{\gamma\gamma}$. For each salt concentration, the viscosity was measured over more than five independent simulation runs of length 5 ns with the stress tensor sampled every integration time step (1 fs).

The Na$^+$ diffusion coefficients were extracted from the long-time limit of the ion's mean-square displacement:

$$D = \lim_{t\to\infty} \frac{1}{6 N_p t} \sum_{i=1}^{N_p} \left\langle \left[ \mathbf{r}_i(t) - \mathbf{r}_i(0) \right]^2 \right\rangle, \qquad (5)$$

where $N_p$ is the number of sodium ions in the system, $\mathbf{r}_i(t)$ is the position of the $i$-th ion at time $t$, and the brackets $\langle \cdots \rangle$ stand for ensemble averaging. The obtained diffusion coefficients were

corrected for finite-size effects using the Yeh-Hummer relation[87]:

$$D_{\infty} = D + \frac{k_{B}T\xi}{6\pi\eta L} \tag{6}$$

with the diffusion coefficient $D_{\infty}$ corresponding to a macroscopic system, $D$ being obtained in a cubic simulation box with side length $L$, and $\xi \approx 2.837297$. The calculated values of viscosity $\eta$ in Eq. (3) were used for evaluating $D_{\infty}$ in Eq. (6). The finite-size correction term corresponded to 17–22% of the measured value $D$.

## Data availability
The data generated and/or analyzed in this study are provided in the Source Data file and are available from the corresponding author on request. Source data are provided with this paper.

## Code availability
The MetalWalls[79] code used for this study is available open source at GitLab (https://gitlab.com/ampere2/metalwalls).

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

## Acknowledgements

We would like to thank Dr. Seena Dehkharghani for the thoughtful discussions on the influence of temperature on water dynamics and NMR relaxation parameters and Dr. Antoine Carof for useful discussions on the modeling of quadrupolar NMR relaxation. This work was supported in part by the National Institutes of Health (NIH): grant no. R01EB026456 (G.M.). This project received funding from the European Research Council under the European Union's Horizon 2020 research and innovation program (grant agreement no. 863473, B.R.). The authors acknowledge access to HPC resources from GENCI (grant no. A0110912966, B.R.), and A.J. wishes to acknowledge the HPC resources of NYU. A.J. acknowledges funding from the US National Science Foundation under award no. CHE2108205.

## Author contributions

L.A., B.R., A.J., and I.C. designed the research with the contributions from E.V.S. and G.M. E.V.S. performed the experiments. I.C. carried out the simulations and data analysis. All authors interpreted the results. I.C. wrote the paper with the contributions of L.A., E.V.S., G.M., A.J., and B.R.

## Competing interests

The authors declare no competing interests.
