## [Peer Review File · Nature Communications]

REVIEWER COMMENTS

Reviewer #1 (Remarks to the Author):

I think the article is generally well-written and with very good technical quality. However, in the current form it does not bring out that breakthrough that would be needed to be considered in Nature Communications? It is a nice study that combines NMR relaxation measurements with MD to explain relaxation in quadrupolar nuclei, which is fine, but is not clear how novel this is and why this is a key result for the field. I think this aspect should be better addressed. If this aspect is addressed, then the paper can be considered for publication.

Other comments below:

- The sentence: "The NMR rate $1/T_1$ grows with increasing salt concentration c and with reducing temperature T (Fig. 3), suggesting that the slowing down of EFG fluctuations determines the rate behavior (Fig. 2)" is a bit confusing. What does it mean a slow down of EFG in this context?

- Related to the previous point, an increasing salt concentration and reducing temperature leads to a higher viscosity of the sample, hence lower mobility. For $1/2$ spins the T_1 vs correlation time will first decrease, reach a minimum, then increase. So one could be either side of the minimum, in which case T_1 could either decrease or increase with a slowing down of molecular dynamics.

- I find the following sentence of the conclusions vague: "the inherently collective character of the EFG at the ion position"

Reviewer #2 (Remarks to the Author):

Manuscript by Chubak et al. describes modelling of quadrupolar ^{23}Na NMR relaxation in aqueous NaCl, NaBr and NaF samples. The authors use DFT calculations to parametrize quadrupolar coupling constant (QCC) and classical MD simulations to probe long-time electric field gradient (EFG) fluctuations at various concentrations and temperatures. They can reproduce experimental relaxation rates accurately especially at lower salt concentrations. The modeling shows that the temperature dependence of T_1 relaxation rate is dominated by the changes in the rate of EFG fluctuations, whereas changes in QCC are minor. The authors show that conventional rotational models based on water dipole reorientation or Stokes-Einstein-Debye relation overestimate the effective correlation time of EFG fluctuations at least an order of magnitude. They conclude that the conventional models oversimplify the inherently collective character of the EFG at the ion position. The multiscale methodology introduced in the manuscript increases significantly understanding quadrupolar ^{23}Na relaxation and time scales of coherent molecular motion in aqueous salt systems, and it provides a new tool for studying dynamics of ions in many interesting applications related to electrolytic systems. The manuscript is very well written, and theoretical background is accurately and thoroughly described.

Minor suggestions:

Term "ultrafast" is used in title only. I suggest clarifying it in the manuscript or replacing it by "fast".

Define abbreviation ACF.

Page 4, left column, second last paragraph: "As seen in Eq. (1) in Methods, the quadrupolar rate..." -> "As seen in Eq. (1) in Methods, the quadrupolar relaxation rate".

Page 6, left column, " r_0 assumes values between 1.5 and 2.0 Å for considered parameters

(Supplementary Fig. S17)”: period after the sentence missing.

Conclusions and perspectives: consider highlighting increasing interest in sodium-ion batteries as a possible complementary technology to lithium-ion batteries.

Reviewer #3 (Remarks to the Author):

Comment on the ms

Quadrupolar $^{23}\text{Na}^+$ NMR Relaxation as a Probe of Ultrafast Collective Dynamics in Aqueous Electrolyte Solutions

by Chubak, Alon, Silletta, Madelin, Jerschow, Rotenberg

The paper is a simulation work which combines ab initio calculations and classical molecular dynamics to account for electron cloud effects and long-time fluctuations in order to calculate the quadrupolar ^{23}Na NMR autocorrelation function (ACF; by the way, the acronym is not introduced) of sodium ions in dilute and concentrated sodium aqueous electrolytes. The classical Sternheimer approximation to calculate the electric field gradient (EFG), needed for calculating absolute NMR relaxation rate, is found to be inadequate and is improved. The classical MD simulations of the ACF reveal a fast short-time decay and a stretched long-time decay, both, well-known features of liquid dynamics. The long-time tail of the ACFs claimed to indicate a hydrodynamic power law is not convincingly documented, I think. From integrating the ACF and the corrected EFG, ^{23}Na relaxation rates are derived as a function of temperature and concentration which well agree with those from experimental ^{23}Na relaxation, in particular, at low concentration. Using further information from the MD trajectories (like viscosity and diffusion coefficient) the microscopic parameters of the relaxation and the application of relaxation models are discussed. For example, the Stokes-Einstein-Debye relation is not found to apply, again a finding known for many associated liquids.

All in all, this an extensive simulation study which reveals many details of the dynamics in electrolytes, although the findings do not present a fully new picture and do not cover times beyond 100ps. I have to note that I am an experimentalist, and fully assessing the MD work is beyond my expertise. My major concern is that the semi-quantitative agreement with experimental ^{23}Na relaxation does not tell much beyond the agreement of the time constant of experiment and simulation. As the dynamics is in the extreme narrowing limit, NMR relaxation does not provide any information on the shape of the ACF. Thus, I find the emphasis in title and abstract as well as in the last paragraph somewhat misleading. By the way, I also do not recognize “ultra fast” dynamics. Most MD simulations of liquids display a sub-ps decay. And “collective” I would apply for sodium-sodium correlations. Are they included in the simulation? Rectifying these issues (see also below), I think, the manuscript may deserve publication.

Further comments

That the dipolar coupling among water protons and sodium ions can be ignored, needs clear justification.

Reply to Reviewer 1

Comments:

1) I think the article is generally well-written and with very good technical quality. However, in the current form it does not bring out that breakthrough that would be needed to be considered in Nature Communications? It is a nice study that combines NMR relaxation measurements with MD to explain relaxation in quadrupolar nuclei, which is fine, but is not clear how novel this is and why this is a key result for the field. I think this aspect should be better addressed. If this aspect is addressed, then the paper can be considered for publication.

Reply: We thank the Reviewer for their positive feedback and their useful suggestions. The context of our work is the study of electrolyte dynamics, the confirmation of experimentally measured trends by computation, and the establishment of dynamic models associated with changes in ion surroundings. We have specifically established the following points: (i) Quadrupolar relaxation as measured with NMR spectroscopy is sensitive to ultrafast (sub-picosecond dynamics), thereby establishing new observables for the study of electrolytes; (ii) refuting previous models for quadrupolar relaxation based on water reorientation or cluster rotation, by proving that dynamics in the fluctuating interactions happen at least an order of magnitude more rapidly than those mechanisms; (iii) demonstrating that the first two ion solvation shells are dominant in determining the experimentally observed trends, and (iv) that in the concentrated salt solutions examined, the co-ion and counter-ion effects mostly affect water retardation globally. All this is achieved with a new computational methodology for calculating quadrupolar relaxation from molecular dynamics trajectories, by combining ab-initio-benchmarked classical MD quantities. We therefore view the work to expand on our ability to assess electrolyte systems beyond infinite dilution, and to provide important methodological advances. We note also that agreement with experimental data is exceptional given the fact that it is extremely challenging to model concentrated electrolyte solutions. The results of this work are relevant to electrolyte dynamics, for example, in the context of rechargeable batteries, but also have significance and implications for the use of ^{23}Na MRI in vivo, where one can actually produce image contrast that is sensitive to the quadrupolar interaction.

To reflect these points better, we would like to highlight several specific key advances of this work and the modifications that have been made as highlighted with red below:

A) The interpretation of quadrupolar relaxation mechanisms is challenging both in experiments and simulations. On the one hand, the quadrupolar coupling constants (QCC) cannot be probed directly for the respective nuclei in solution, which limits the extraction of specific correlation times and thus hinders a quantitative analysis of relaxation mechanisms. On the other hand, both state-of-state ab initio and classical simulations (e.g., Refs. [32, 39, 42] in the manuscript) featured considerable uncertainties/discrepancies in the computed rates and were mostly constrained to the case of infinite ion dilution, thereby again limiting a quantitative and systematic discussion concerning the relaxation mechanisms. In this work, we apply a combination of experiments and different simulation techniques to (i) achieve a quantitative agreement with experiments, which enabled us (ii) a systematic discussion on the relaxation mechanisms.

We have emphasized the latter points on p. 2 of the Introduction and p. 7 of the Conclusions of the revised manuscript:

Hence, uncertainties arising both in ab initio and classical MD based approaches hindered the quantitative comparison with experimental NMR rates and a systematic analysis of the quadrupolar relaxation mechanisms.

The availability of dynamic information over a broad range of system parameters enabled us to have a consistent discussion concerning the quadrupolar relaxation models.

B) We provide a thorough assessment of the common Sternheimer approximation for ^{23}Na over a broad range of salt concentrations and we show that it is not adequate for a quantitative description of the ^{23}Na QCC. We offer a strategy for the parametrization of the latter that allows us reaching a good agreement with the experiments.

C) The use of classical MD to sample long-time fluctuations allowed us to accurately estimate the EFG correlation times for a broad range of system parameters. Our simulation reveal several hitherto unobserved features of the EFG ACFs, such as (i) non-exponential decays for ~ 1 – 10 ps and (ii) signatures of a power-law decay at longer times (please also see Questions 2 and 3 of Reviewer 3 for a more detailed discussion). We have emphasized the latter points on p. 3 of the revised manuscript:

We provide a systematic description of the $^{23}\text{Na}^+$ EFG relaxation at short, intermediate, and long times ranging from a few fs to tens of ps.

D) The good agreement with the experiments and availability of the relaxation data over a broad range of system parameters allowed a consistent and quantitative discussion on the relaxation mechanisms (such as, the relation to single water molecule reorientation and the parametrization using the Stokes-Einstein-Debye picture). To the best of our knowledge, such rotational models have not been assessed in detail, despite being commonly used to estimate the correlation times to date (e.g., Refs. [20-21] in the manuscript). We show that such models provide a significant overestimation of the measure EFG correlation times, highlighting that the quadrupolar relaxometry does not serve as a direct probe of the single ion diffusion or single water molecule reorientation dynamics. Please also see the discussion in the Question 4 of Reviewer 3.

E) We highlight the the quadrupolar relaxation is (i) *ultrafast* (that is, on the sub-ps time scale); continuous-solvent or single water molecule reorientation pictures yield τ_c in the range between several and tens of ps; (ii) *collective*, as caused by a combination of molecular mechanisms; the latter is evidenced by the stretched-exponential shape of the ACFs as well as a comparison with other relaxation functions (in particular, that of the stress tensor); (iii) *local*, as largely determined by the dynamics in the two solvation shells around the ion. To support this we provide an additional Supplementary Figure S8 in the SI comparing the EFG ACFs from the point charges within the first solvation shell and the first two solvation shells against the full ACF over a range of salt concentrations. While the first solvation shell contribution displays a generally slower relaxation dynamics, the EFG ACFs are captured quantitatively well by the dynamics of charges within the two solvation shells at all concentrations considered.

In the revised SI, we have included a Supplementary Figure S8 and provide an additional discussion on p. 9:

In this section, we discuss the origin of the EFG fluctuations in terms of the microscopic environment of the Na^+ ion. In Supplementary Fig. S8, we show the EFG ACFs at $T = 25^\circ\text{C}$ and multiple concentrations, as computed from all point charges in the system using Ewald summation, from water molecules and ions residing in the first solvation shell only, and from water molecules and ions within the first two solvation shells. We find that the EFG ACF due to the first solvation shell decays much more slowly than the full EFG ACF, in particular with increasing salt concentration. In contrast, the EFG ACF due to point charges in the first two solvation shells provides a good description of the relaxation with the corresponding ACF capturing well both the short- and long-time dynamics (compare blue and black lines in Fig. S8). This indicates that the processes in the first two solvation shells are mainly responsible for the quadrupolar relaxation dynamics.

We also provide additional statement on p. 5 of the revised manuscript:

Similarly, we find that the EFG relaxation dynamics is well captured by the point charge contributions located within the first two solvation shells around the ion, whereas the EFG due to the first solvation shell relaxes much more slowly (Supplementary Fig. S8).

In summary, rather than directly probing the single ion diffusion or single water molecule reorientation dynamics, the quadrupolar $^{23}\text{Na}^+$ relaxation senses the fast collective dynamics around the solute. Upon a proper interpretation of the quadrupolar relaxation, a complementary view on such dynamics can be obtained from NMR experiments, in addition to nanoscale rheology, solvation dynamics, or THz dielectric spectroscopy. For the latter, quadrupolar NMR could help resolve and understand secondary Debye peaks at ≈ 0.3 THz that are typically obtained through fits with many parameters [Balos, V., Imoto, S., Netz, R. R., Bonn, M., Bonthuis, D. J., Nagata, Y., and Hunger, J. Macroscopic conductivity of aqueous electrolyte solutions scales with ultrafast microscopic ion motions. Nat. Commun, 11(1), 1-8 (2020)].

We have accordingly modified our discussion on p. 7-8 of the revised Conclusions:

Rather than directly probing single ion diffusion or single water molecule reorientation, our results suggest that the quadrupolar NMR relaxometry of $^{23}\text{Na}^+$ may be used as a complementary tool to analyze electrolyte dynamics in the THz domain, as relevant for emerging sodium-ion battery technologies [9-11]. As the quadrupolar relaxation is largely determined by the processes in the immediate vicinity of the solute, it can provide supplementary information on the fast, collective, molecular motions in ionic solvation cages that have been associated with the high-frequency dielectric response [63] solvation dynamics [64], as well as structural relaxation [61] in aqueous electrolytes.

Finally, to contextualize better the content and advances our work we have modified the discussion on p. 2 of the Introduction and the Abstract of the revised manuscript:

Here we show that applying ab initio calculations to parametrize C_Q in conjunction with classical MD to evaluate τ_c allows reaching a good agreement between the calculated and experimentally-obtained quadrupolar rates of $^{23}\text{Na}^+$ in electrolyte solutions over a broad range of salt concentrations and temperatures, thereby enabling a systematic analysis of the relaxation pathways and detailed modeling of the underlying dynamics. We find that the main effect of increased relaxivity is due to a lengthening of the correlation times, rather than a change of the average quadrupolar coupling constant. Counterintuitively, the latter varies mildly over the range of considered parameters, slightly decreasing with concentration and increasing with temperature. We conclude that, contrary to the commonly-assumed picture, rotational models based on the water dipole reorientation or Stokes-Einstein-Debye relation significantly overestimate the EFG correlation times. Rather, our results indicate that the EFG relaxation is mainly determined by the dynamics in the first two solvation shells around the solute and occurs over a time scale comparable to that of solution structural rearrangements. This work thus suggests that the ultrafast (subpicosecond) collective dynamics of the liquid primarily drives the quadrupolar relaxation at the sodium ion position, thereby providing new guidelines for the interpretation of quadrupolar relaxation rates in electrolyte systems.

Nuclear magnetic resonance relaxometry represents a powerful tool for extracting dynamic information. Yet, obtaining links to molecular motion is challenging for many ions that relax through the quadrupolar mechanism, which is mediated by electric field gradient fluctuations and lacks a detailed microscopic description. For sodium ions in aqueous electrolytes, we combine ab initio calculations to account for electron cloud effects with classical molecular dynamics to sample long-time fluctuations, and obtain relaxation rates in excellent agreement with experiments over broad concentration and temperature ranges. We demonstrate that quadrupolar nuclear relaxation is sensitive to ultrafast (subpicosecond) dynamics not captured by previous models based on water reorientation or cluster rotation. While ions affect the overall water retardation, the experimental trends are mainly explained by dynamics in the first two solvation shells of sodium, which contain mostly water. This work thus provides a new basis for studying quadrupolar relaxometry in electrolyte or bioelectrolyte systems.

Other comments below:

2) The sentence: “The NMR rate $1/T_1$ grows with increasing salt concentration c and with reducing temperature T (Fig. 3), suggesting that the slowing down of EFG fluctuations determines the rate behavior (Fig. 2)” is a bit confusing. What does it mean a slow down of EFG in this context?

Reply: In this context, the slow down of EFG fluctuations with increasing the salt concentration or decreasing temperature results in a slower decay of the EFG autocorrelation function and, accordingly, an increased correlation time τ_c . In fact, we show in more detail in Fig. 4, that the rate behavior is mainly dictated by changes in τ_c rather than in the quadrupolar coupling constant. We have clarified the statement on p. 4 (see below in the reply to the following comment).

3) Related to the previous point, an increasing salt concentration and reducing temperature leads to a higher viscosity of the sample, hence lower mobility. For 1/2 spins the T_1 vs correlation time will first decrease, reach a minimum, then increase. So one could be either side of the minimum, in which case T_1 could either decrease or increase with a slowing down of molecular dynamics.

Reply: We thank the Reviewer for pointing this out. Indeed, the reversal of the longitudinal rate dependence on τ_c for 1/2 spins happens for sufficiently low molecular mobility (i.e., sufficiently high τ_c). This crossover occurs for τ_c between ~ 100 ps and ~ 1 ns for typical ^1H NMR relaxation parameters [e.g., Fig. 2.6 in Kowalewski, J. and Mäler, L. Nuclear spin relaxation in liquids: theory, experiments, and applications. CRC press, (2017)]. As we discuss in more detail in Methods, in the case of sodium, the inverse Larmor frequency $\omega_0^{-1} \approx 7.6$ ns is much larger than the typical observed EFG correlation times τ_c (~ 1 ps and below). Thus, it is safe to assume that the extreme narrowing condition, under which $1/T_1 \propto \tau_c$, holds well for the considered concentration and temperature range.

We have made the following changes in the paragraph on p. 4 of the revised manuscript:

The NMR rate $1/T_1$ grows with increasing salt concentration c and with reducing temperature T (Fig. 3). Under the extreme narrowing condition, which is fulfilled for the considered cases (see Methods), $1/T_1 \propto \tau_c$, thereby suggesting that the slowing down of EFG fluctuations (Fig. 2), as reflected in the augmented correlation time, determines the

rate behavior.

4) I find the following sentence of the conclusions vague: “the inherently collective character of the EFG at the ion position.”

Reply: In the present case, the source of the EFG at the ion position \mathbf{r}_I is intermolecular and due to charges in its surrounding environment. If $V_{\alpha\beta}(\mathbf{r}_i, \mathbf{r}_I)$ is the EFG at \mathbf{r}_I due to a charge q_i at \mathbf{r}_i , then the full EFG at \mathbf{r}_I due to all charges in the system is

$$V_{\alpha\beta}(\mathbf{r}_I) = \sum_i V_{\alpha\beta}(\mathbf{r}_i, \mathbf{r}_I). \quad (1)$$

The EFG ACF at \mathbf{r}_I can then be expressed as

$$\begin{aligned} C_{\text{EFG}}(t, \mathbf{r}_I) &= \left\langle \sum_{\alpha\beta} V_{\alpha\beta}(t, \mathbf{r}_I) V_{\alpha\beta}(0, \mathbf{r}_I) \right\rangle \\ &= \left\langle \sum_{\alpha\beta} \left(\sum_i V_{\alpha\beta}(t, \mathbf{r}_i, \mathbf{r}_I) V_{\alpha\beta}(0, \mathbf{r}_i, \mathbf{r}_I) + \sum_{i,j \neq i} V_{\alpha\beta}(t, \mathbf{r}_i, \mathbf{r}_I) V_{\alpha\beta}(0, \mathbf{r}_j, \mathbf{r}_I) \right) \right\rangle. \end{aligned} \quad (2)$$

We note that the expression above is a function of either the positions of two ($\mathbf{r}_i, \mathbf{r}_I$) or three ($\mathbf{r}_i, \mathbf{r}_j, \mathbf{r}_I$) distinct particles, highlighting its collective character and a dependence on multi-body correlations.

We have modified the sentence on p. 7 of the conclusions to be more precise:

This disagreement is understandable as these models restrict the relaxation description to one- or two-body correlations, **oversimplifying the collective dynamics of the intermolecular EFG at the ion position.**

Reply to Reviewer 2

Comments:

1) Manuscript by Chubak et al. describes modelling of quadrupolar ^{23}Na NMR relaxation in aqueous NaCl, NaBr and NaF samples. The authors use DFT calculations to parametrize quadrupolar coupling constant (QCC) and classical MD simulations to probe long-time electric field gradient (EFG) fluctuations at various concentrations and temperatures. They can reproduce experimental relaxation rates accurately especially at lower salt concentrations. The modeling shows that the temperature dependence of T_1 relaxation rate is dominated by the changes in the rate of EFG fluctuations, whereas changes in QCC are minor. The authors show that conventional rotational models based on water dipole reorientation or Stokes-Einstein-Debye relation overestimate the effective correlation time of EFG fluctuations at least an order of magnitude. They conclude that the conventional models oversimplify the inherently collective character of the EFG at the ion position. The multiscale methodology introduced in the manuscript increases significantly understanding quadrupolar ^{23}Na relaxation and time scales of coherent molecular motion in aqueous salt systems, and it provides a new tool for studying dynamics of ions in many interesting applications related to electrolytic systems. The manuscript is very well written, and theoretical background is accurately and thoroughly described.

Reply: We thank the Reviewer for their positive feedback on the initial version of our manuscript and suggestions to improve it.

Minor suggestions:

2) Term “ultrafast” is used in title only. I suggest clarifying it in the manuscript or replacing it by “fast”.

Reply: We thank the Reviewer for raising this point and we agree that we should have been more specific with the definition. By the “ultrafast dynamics”, we meant subpicosecond-scale motions that can also be probed with dielectric spectroscopy in the THz domain (e.g., some the recent works used this term when referring to such dynamics [Balos, V., Imoto, S., Netz, R. R., Bonn, M., Bonthuis, D. J., Nagata, Y., and Hunger, J. Macroscopic conductivity of aqueous electrolyte solutions scales with ultrafast microscopic ion motions. Nat. Commun, 11(1), 1-8 (2020)]). We have clarified this on p. 2 of the revised manuscript and in the abstract:

This work thus suggests that the ultrafast (subpicosecond) collective dynamics of the liquid primarily drives the quadrupolar relaxation at the sodium ion position, thereby providing new guidelines for the interpretation of quadrupolar relaxation rates in electrolyte systems.

We demonstrate that quadrupolar nuclear relaxation is sensitive to ultrafast (subpicosecond) dynamics not captured by previous models based on water reorientation or cluster rotation.

3) Define abbreviation ACF.

Reply: We thank you for pointing this out. We have now defined the term on p. 3 of the revised manuscript:

Fig. 2a shows the autocorrelation functions (ACFs) of the classical EFG at the Na^+ position, ...

4) Page 4, left column, second last paragraph: “As seen in Eq. (1) in Methods, the quadrupolar rate...” → “As seen in Eq. (1) in Methods, the quadrupolar relaxation rate.”

Reply: We have followed the suggestion and modified the sentence accordingly on p. 4 on the revised manuscript.

5) Page 6, left column, “ r_0 assumes values between 1.5 and 2.0 Å for considered parameters (Supplementary Fig. S17)”: period after the sentence missing.

Reply: We have added the missing period at the end of this sentence.

6) Conclusions and perspectives: consider highlighting increasing interest in sodium-ion batteries as a possible complementary technology to lithium-ion batteries.

Reply: We thank the Reviewer for this suggestions and we have highlighted this on p. 7 of the revised conclusions:

Our results suggest that the quadrupolar NMR relaxometry of $^{23}\text{Na}^+$ may be used as a complementary tool to analyze electrolyte dynamics in the THz domain, **as relevant for emerging sodium-ion battery technologies.**

Reply to Reviewer 3

Comments:

1) The paper is a simulation work which combines *ab initio* calculations and classical molecular dynamics to account for electron cloud effects and long-time fluctuations in order to calculate the quadrupolar ^{23}Na NMR autocorrelation function (ACF; by the way, the acronym is not introduced) of sodium ions in dilute and concentrated sodium aqueous electrolytes. The classical Sternheimer approximation to calculate the electric field gradient (EFG), needed for calculating absolute NMR relaxation rate, is found to be inadequate and is improved.

Reply: We thank the Reviewer for their positive feedback on the initial version of our manuscript and suggestions to improve it. In addition, we would like to emphasize that we do not only combine *ab initio* calculations with classical molecular dynamics simulations, but also with NMR experiments, and find quantitative and semi-quantitative agreement (e.g., for the comparison between the sodium relaxation rates and diffusion coefficients) between the theoretical and experimental approaches. In particular, our discussion on the mechanisms behind the quadrupolar ^{23}Na relaxation in Fig. 5 relies on the combination of the data from simulations and experiments. Finally, we have now introduced the missing definition of the acronym on p. 3 of the revised manuscript:

Fig. 2a shows the autocorrelation functions (ACFs) of the classical EFG at the Na^+ position, ...

2) The classical MD simulations of the ACF reveal a fast short-time decay and a stretched long-time decay, both, well-known features of liquid dynamics.

Reply: Indeed, the combination of a fast relaxation at short times and a slow one at long times is a known feature of liquid dynamics, and it has been probed with other experimental techniques (e.g., fluorescence experiments that give information about the solvation dynamics [Jimenez, R., Fleming, G. R., Kumar, P. V., and Maroncelli, M. Femtosecond solvation dynamics of water. *Nature*, 369(6480), 471-473 (1994)]). However, it is not clear how these modes of liquid dynamics are encoded in the EFG fluctuations, and the respective EFG ACF that is probed in the NMR relaxation experiments of quadrupolar nuclei. In fact, the main goal of our work was to combine experiments and computer simulation to gain a better understanding of the EFG relaxation on the microscopic level and to use that information to assess the available and commonly-used models of quadrupolar relaxation.

To the best of our knowledge, it is the first time that a systematic description of the $^{23}\text{Na}^+$ EFG relaxation up to ~ 10 – 100 ps is provided. Most of the previous simulation works (both using state-of-the-art *ab initio* and classical MD approaches, e.g., see Refs. [32, 39, 42] in the manuscript) was limited to the description of relaxation in the ps-regime and to quadrupolar ions at infinite dilution in water. Here, we reveal a non-exponential character of the EFG ACF decay for ~ 1 – 10 ps and also find signatures of the algebraic decay at longer times (see more below), the features that has not been observed before and that point toward a collective (i.e., many-body) mechanism behind the relaxation.

We have emphasized the above-mentioned points on p. 3 on the revised manuscript:

We provide a systematic description of the $^{23}\text{Na}^+$ EFG relaxation at short, intermediate, and long times ranging from a few fs to tens of ps.

3) The long-time tail of the ACFs claimed to indicate a hydrodynamic power law is not convincingly documented, I think.

Reply: We recognize on p. 4 of the manuscript that (i) the behavior of the ACF that is consistent with a $\sim t^{-5/2}$ algebraic decay is observed over a limited time range; (ii) sampling of EFG fluctuations at longer time scales is necessary to decisively confirm the presence of this algebraic decay. Our sampling of EFG fluctuations was performed over a quite long time period (25 ns), and extending it considerably to obtain accurate ACFs beyond ~ 100 ps would require too much computational resources. We believe that the application of mesoscopic electrokinetic methods that can probe much longer time scales may be useful for decisively verifying the presence of this hydrodynamic tail over multiple decades in time, and we leave it for future work. In addition, here we summarize the findings that support the presence of the algebraic relaxation at long times: (i) it is observed systematically for all concentrations and temperatures considered in our simulations; (ii) the multi-exponential and stretched-exponential fits for the ACFs decay systematically slower than the MD data at long times (Supplementary Figs 10 and 11); (iii) in the Supplementary Sec. II A, we provide an analysis for finite-size effects in our simulations that can potentially affect

hydrodynamic power-law regimes of the measured ACFs [Asta, A. J., Levesque, M., Vuilleumier, R. and Rotenberg, B. Transient hydrodynamic finite-size effects in simulations under periodic boundary conditions. Phys. Rev. E 95, 061301 (2017)]. We do not observe a finite-box size effect on our EFG ACFs; (iv) $\sim t^{-5/2}$ relaxation of the EFG ACF at long times was predicted by the mode coupling theory of Bosse et al. [Bosse, J., Quitmann, D. and Wetzel, C. Mode-coupling theory of field-gradient correlation functions: The quadrupolar relaxation rate in liquids. Phys. Rev. A 28, 2459–2473 (1983)]; (v) in the revised SI, we have included a new Fig. S14 that shows the EFG ACFs $C_{\text{EFG}}(t)$ multiplied by $t^{5/2}$ and $t^{3/2}$ at $c = 4$ m and different temperatures. We find that $t^{5/2}C_{\text{EFG}}(t)$ saturates at a constant value at long times, supporting the presence of an algebraic $\sim t^{-5/2}$ decay.

In the revised SI, we have thus included a new Figure S15, and added a complementary discussion on p. 11:

In Fig. S14, we additionally highlight the long-time behavior of the EFG ACFs by multiplying it by $t^{5/2}$ and $t^{3/2}$. As seen in Fig. S14a, $t^{5/2}C_{\text{EFG}}(t)$ saturates at long times, suggesting an algebraic decay $\sim t^{-5/2}$ that is consistent with the mode coupling theory of Bosse et al. [19].

We have updated the corresponding discussion on p. 4 of the revised manuscript:

Although observed over a limited time range (up to a decade), we find that the long-time tail of the EFG ACFs is consistent with a power law $\sim t^{-5/2}$, as shown with $C_{\text{EFG}}(t)$ on a log-log scale for $c = 4$ m in Fig. 2c and with $t^{5/2}C_{\text{EFG}}(t)$ in Supplementary Fig. S15.

We have also updated the caption of Figure 2 on p. 3 of the revised manuscript:

See also Supplementary Fig. S15 for $C_{\text{EFG}}(t)$ multiplied by $t^{5/2}$ and $t^{3/2}$.

4) From integrating the ACF and the corrected EFG, ^{23}Na relaxation rates are derived as a function of temperature and concentration which well agree with those from experimental ^{23}Na relaxation, in particular, at low concentration. Using further information from the MD trajectories (like viscosity and diffusion coefficient) the microscopic parameters of the relaxation and the application of relaxation models are discussed. For example, the Stokes-Einstein-Debye relation is not found to apply, again a finding known for many associated liquids.

Reply: We agree with the Reviewer that the Stokes-Einstein-Debye (SED) relation is rooted in a Brownian rotational diffusion mechanism and is not expected to hold at the molecular scale, as we acknowledge on p. 5 of the manuscript. However, the quality of the SED model and, in general, the applicability of a rotational mechanism to quadrupolar relaxation has not been systematically assessed, despite being commonly used to date (e.g., see Refs. [20–21] in the manuscript). One of the main difficulties that prevented the assessment of those models was (i) a lacking knowledge of the quadrupolar coupling constants in solution that would allow to compute the EFG correlation times directly from the experiments; (ii) lack of quantitative agreement between the experimental and simulated $^{23}\text{Na}^+$ quadrupolar rates even in recent, state-of-the-art *ab initio* or classical MD simulations (e.g., Refs. [32] and [42] in the manuscript); (iii) typical availability of the $^{23}\text{Na}^+$ relaxation data for the infinite dilution only (thus, impeding the assessment of trends with temperature or solution viscosity). In this work, by improving upon the Sternheimer approximation and considering aqueous solutions over a broad range of concentrations and temperatures, we have managed to overcome the above-mentioned issues, enabling us a quantitative assessment of the relaxation mechanisms. In particular, our results show that the parametrization of τ_c based on the translational Stokes radius of $^{23}\text{Na}^+$ leads to a significant overestimation of the correlation time. We further highlight the breakdown of an important hypothesis (separation of time scales) underlying continuous-solvent models (in particular, the SED one), as the relaxation of EFG fluctuation occurs over time scales comparable to structural rearrangements in the solution. Finally, we note, in particular the following non-intuitive findings: the average EFG decreases slightly with increasing concentration, and increases with increasing temperature. In addition, the major experimental trends are explained by the dynamics in the first two solvation shells. None of these findings have been known or could have been guessed based on prior theory or experiment.

We have additionally highlighted the above-mentioned points on p. 2 of revised version of the manuscript:

Hence, uncertainties arising both in *ab initio* and classical MD based approaches hindered the quantitative comparison with experimental NMR rates and a systematic analysis of the quadrupolar relaxation mechanisms.

We also provide an update discussion in the Introduction and Conclusions, as detailed in the response of the Question 1 of the Reviewer 1.

5) All in all, this an extensive simulation study which reveals many details of the dynamics in electrolytes, although the findings do not present a fully new picture and do not cover times beyond 100 ps. I have to note that I am an experimentalist, and fully assessing the MD work is beyond my expertise. My major concern is that the semi-quantitative agreement with experimental ^{23}Na relaxation does not tell much beyond the agreement of the time constant of experiment and simulation. As the dynamics is in the extreme narrowing limit, NMR relaxation does not provide any information on the shape of the ACF.

Reply: We note that the computational results show that the correlation functions can be considered fully decayed well within 100 ps (decay over at least two orders of magnitude), and the statistics of MD can be considered fully converged, as seen from the running integrals of the EFG ACFs in Supplementary Fig. S16.

We agree that experimentally, the fast motion regime applies. This also allows to directly relate the measured relaxation rate $1/T_1$ to the correlation time τ_c , that is $1/T_1 \propto \tau_c$. We indicate this on p. 4 of the revised manuscript:

Under the extreme narrowing condition, which is fulfilled for the considered cases (see Methods), $1/T_1 \propto \tau_c$, thereby suggesting that the slowing down of EFG fluctuations (Fig. 2), as reflected in the augmented correlation time, determines the rate behavior.

The agreement of the time constant in experiments and simulations allows us to have a quantitative discussion concerning the mechanisms of the quadrupolar relaxation. To the best of our knowledge, the latter aspect has not been addressed in detail in other studies due to (i) significant uncertainties on the previously-computed NMR rates in simulations; (ii) lack of a systematic exploration of the effects of temperature and concentration. Please note also our responses to Reviewer 1, which highlight the information we are able to obtain from this comparison between computation and experiment.

6) Thus, I find the emphasis in title and abstract as well as in the last paragraph somewhat misleading. By the way, I also do not recognize “ultra fast” dynamics. Most MD simulations of liquids display a sub-ps decay. And “collective” I would apply for sodium-sodium correlations. Are they included in the simulation?

Reply: We acknowledge the fact that the use of the term “ultrafast” might be somewhat misleading in this context. By the “ultrafast dynamics”, we meant subpicosecond-scale motions that can also be probed with dielectric spectroscopy in the THz domain (e.g., [Balos, V., Imoto, S., Netz, R. R., Bonn, M., Bonthuis, D. J., Nagata, Y., and Hunger, J. Macroscopic conductivity of aqueous electrolyte solutions scales with ultrafast microscopic ion motions. Nat. Commun, 11(1), 1-8 (2020)]). We have clarified this on p. 2 of the revised manuscript and in the abstract:

This work thus suggests that the ultrafast (subpicosecond) collective dynamics of the liquid primarily drives the quadrupolar relaxation at the sodium ion position, thereby providing new guidelines for the interpretation of quadrupolar relaxation rates in electrolyte systems.

We demonstrate that quadrupolar nuclear relaxation is sensitive to ultrafast (subpicosecond) dynamics not captured by previous models based on water reorientation or cluster rotation.

In our simulations, sodium-sodium correlations are included, as both water molecules and ions contribute to the EFG at the ion position. We provide an additional analysis concerning the effect of ions on the EFG relaxation. On p. 5 of the SI, we show that even at the highest $c = 4$ m considered, there are on average around 20.6 oxygens within the first two solvation shells of sodium, as compared to 0.8 other sodium cations and 1.9 chloride anions. At all concentrations, the first solvation shells is predominantly composed of water molecules. As the full EFG relaxation is captured well by the dynamics within the first two solvation shells (new Supplementary Fig. S8), the latter observations indicate that other ions in the system enter into the coordination shell of Na^+ , yet they are much less populous than the solvent and are also further away in general (see radial distribution functions in Supplementary Figs. S3 and S4). This suggests that water provides the dominant contribution to the EFG within the given concentration range, whereas ions act mainly to slow down water dynamics. We have commented on this on p. 5 of the revised manuscript:

The first two solvation shells of Na^+ are predominantly populated by water molecules even at the highest $c = 4$ m considered (see Supplementary Sec. IIB), indicating that the solvent provides the main contribution to the EFG at the Na^+ position and suggesting that other ions mainly retard water dynamics.

We have also clarified the discussion on p. 8 of the manuscript:

Future work could also focus on developing mesoscopic approaches that would allow a quantitative description of quadrupolar relaxation in complex biological-type compartments, characterized either by slow-motion conditions with dynamics within intracellular and extracellular spaces in biological tissues that may include structural anisotropy with residual quadrupolar coupling and a distribution of correlation times (as, for example, in **connective tissue** where sodium ions are surrounded by a collagen matrix [4]). **Such models would further provide a foundation for the interpretation of magnetic resonance imaging contrast mechanisms that are sensitive to the quadrupolar interaction [67-70]).**

Finally, we have modified the abstract to better highlight the content and advances of our work:

Nuclear magnetic resonance relaxometry represents a powerful tool for extracting dynamic information. Yet, obtaining links to molecular motion is challenging for many ions that relax through the quadrupolar mechanism, which is mediated by electric field gradient fluctuations and lacks a detailed microscopic description. For sodium ions in aqueous electrolytes, we combine ab initio calculations to account for electron cloud effects with classical molecular dynamics to sample long-time fluctuations, and obtain relaxation rates in excellent agreement with experiments over broad concentration and temperature ranges. We demonstrate that quadrupolar nuclear relaxation is sensitive to ultrafast (subpicosecond) dynamics not captured by previous models based on water reorientation or cluster rotation. While ions affect the overall water retardation, the experimental trends are mainly explained by dynamics in the first two solvation shells of sodium, which contain mostly water. This work thus provides a new basis for studying quadrupolar relaxometry in electrolyte or bioelectrolyte systems.

7) Rectifying these issues (see also below), I think, the manuscript may deserve publication.

Reply: We thank again the Reviewer for their useful comments and acknowledging the possible relevance for publication in *Nature Communications*

Further comments:

8) That the dipolar coupling among water protons and sodium ions can be ignored, needs clear justification.

Reply: In the Supplementary Section IC, we provide an estimate of the dipolar contribution to the longitudinal rate $1/T_1^{\text{dd}}$ due to the coupling between water protons and sodium ions. We estimate the latter from the associated correlation function (Eq. (17) in the SI), and the final expression in the extreme narrowing condition is given by

$$\frac{1}{T_1^{\text{dd}}} \approx \frac{3}{2} \left(\frac{\mu_0}{4\pi} \right)^2 \hbar^2 \gamma_I^2 \gamma_H^2 \left\langle \frac{1}{r^6} \right\rangle \tau_c^{\text{dd}} \quad (3)$$

with nuclear gyromagnetic ratios $\gamma_I/2\pi = 11.262 \text{ MHz}\cdot\text{T}^{-1}$ for ^{23}Na and $\gamma_H/2\pi = 42.577 \text{ MHz}\cdot\text{T}^{-1}$ for ^1H . We estimate r from the first peak of the Na-O radial distribution function ($r \approx 2.89 \text{ \AA}$) and assume that the correlation time τ_c^{dd} is around 4 ps, corresponding to the time scale of intermolecular ^1H dipole-dipole relaxation in pure water at room temperature conditions [Singer, P. M., Asthagiri, D., Chapman, W. G. and Hirasaki, G. J. Molecular dynamics simulations of NMR relaxation and diffusion of bulk hydrocarbons and water. *J. Magn. Reson.* 277, 15–24 (2017)]. The final estimate for $1/T_1^{\text{dd}} \approx 2.7 \cdot 10^{-4} \text{ s}^{-1}$ is smaller by more than 4 orders of magnitude than the quadrupolar rate contribution ($\approx 16 \text{ s}^{-1}$) at low salt concentrations and room temperature. Finally, the dipolar contribution is even negligible with much larger estimates for τ_c^{dd} . Thus, we conclude that the quadrupolar interaction dominates the relaxation of $^{23}\text{Na}^+$.

We have rephrased a sentence on p. 4 of the revised manuscript to emphasize this:

Finally, our estimates in Supplementary Note IC for the dipole-dipole contribution to the $^{23}\text{Na}^+$ rate $1/T_1$ due to interactions with the spins of ^1H , ^{23}Na , and ^{35}Cl are more than four orders of magnitude smaller compared to the quadrupolar contribution, indicating that the latter dominates the ^{23}Na NMR relaxation.

REVIEWERS' COMMENTS

Reviewer #1 (Remarks to the Author):

The authors have given reasonable answers to my queries. Before final acceptance, I think a slight update of the literature survey and discussion of the results should be carried out. There has been previous work of Abbott et al. on measuring T1 relaxation on sodium ions in viscous liquids, including in aqueous ones, see Phys. Chem. Chem. Phys., 2016,18, 25528-25537. I think this work is relevant to that reported in this paper.

It would be useful if the authors could compare their T1 relaxation values to those reported in the mentioned paper and perhaps discuss a possible application of their theory to such systems, which are slightly different yet similar in some respects. I think this is important for assessing the relevance of the framework developed by the authors to other systems with sodium ions, as the authors also acknowledge in one of the answers to my comments.

I am happy for the paper to be accepted once such final discussion has been included in the manuscript.

Reviewer #3 (Remarks to the Author):

In the first round the relevant/critical statements of the reviewers were:

"It is a nice study that combines NMR relaxation measurements with MD to explain relaxation in quadrupolar nuclei, which is fine, but is not clear how novel this is and why this is a key result for the field. I think this aspect should be better addressed." (reviewer 1).

Only minor correction, I understand. For example, "Term "ultrafast" is used in title only." I suggest clarifying it in the manuscript or replacing it by "fast"." (reviewer 2)

"All in all, this an extensive simulation study which reveals many details of the dynamics in electrolytes, although the findings do not present a fully new picture and do not cover times beyond 100 ps. . . . My major concern is that the semi-quantitative agreement with experimental ²³Na relaxation does not tell much beyond the agreement of the time constant of experiment and simulation . . . I also do not recognize "ultra fast" dynamics. . . . emphasis in title and abstract as well as in the last paragraph somewhat misleading."(reviewer 3).

The authors have now written a ten pages long rebuttal indicating many changes (and offering a lot of discussions).

Still, I am not fully convinced about the novelty of the results (reviewer 1 and 3). Actually, not much is changed with respect to this issue. Also, the reply addressing the contribution of NMR relaxation is not fully convincing (reviewer 3). For example, I do not agree with the statement (Abstract): "This work thus provides a new basis for studying quadrupolar relaxometry in electrolyte or bioelectrolyte systems." Again, relaxation rates under extreme narrowing conditions are of not much help regarding the shape of the ACF, no "new guidelines" (Introduction) are provided.

The authors insist using the term "ultrafast dynamics". I think the notion "ultrafast" usually does not refer to "picosecond" dynamics.

Reply to Reviewer 1

Comments:

The authors have given reasonable answers to my queries. Before final acceptance, I think a slight update of the literature survey and discussion of the results should be carried out. There has been previous work of Abbott et al. on measuring T1 relaxation on sodium ions in viscous liquids, including in aqueous ones, see Phys. Chem. Chem. Phys., 2016,18, 25528-25537. I think this work is relevant to that reported in this paper.

It would be useful if the authors could compare their T_1 relaxation values to those reported in the mentioned paper and perhaps discuss a possible application of their theory to such systems, which are slightly different yet similar in some respects. I think this is important for assessing the relevance of the framework developed by the authors to other systems with sodium ions, as the authors also acknowledge in one of the answers to my comments.

I am happy for the paper to be accepted once such final discussion has been included in the manuscript.

Reply: We thank the Reviewer for their positive feedback and for the relevant reference that we now cite in the revised manuscript (the new Ref. [66]). The viscosity of salt-glycerol mixtures considered in Phys. Chem. Chem. Phys., 2016, 18, 25528-25537 is much higher than that of aqueous solutions considered in the present work (~ 1000 cP versus ~ 1 cP), and the reported ^{23}Na relaxation times T_1 are more than two orders of magnitude larger compared to the aqueous electrolytes. The latter again reflects the connection between macroscopic behavior (through viscosity) and quadrupolar NMR relaxation. We agree that the approach employed here can be applied to salt-glycerol mixtures, yet more computational resources might be necessary to accurately sample the relevant correlation functions in such highly viscous systems.

We have updated the final Discussion on p. 8 of the revised manuscript:

The ability to capture the NMR relaxation rates by means of classical MD allows elucidating the quadrupolar relaxation mechanisms that occur in multicomponent systems, such as concentrated aqueous solutions of multiple salts [65], mixtures of salts with glycerol [66], or polyelectrolytes [67], where the relaxation dynamics may be influenced by environment heterogeneity, interface formation, microphase separation, or ion binding to polyelectrolyte chains.

Reply to Reviewer 3

Comments:

In the first round the relevant/critical statements of the reviewers were:

“It is a nice study that combines NMR relaxation measurements with MD to explain relaxation in quadrupolar nuclei, which is fine, but is not clear how novel this is and why this is a key result for the field. I think this aspect should be better addressed.” (reviewer 1).

Only minor correction, I understand. For example, “Term “ultrafast” is used in title only. I suggest clarifying it in the manuscript or replacing it by “fast”.” (reviewer 2).

“All in all, this an extensive simulation study which reveals many details of the dynamics in electrolytes, although the findings do not present a fully new picture and do not cover times beyond 100 ps... My major concern is that the semi-quantitative agreement with experimental ^{23}Na relaxation does not tell much beyond the agreement of the time constant of experiment and simulation... I also do not recognize “ultra fast” dynamics; emphasis in title and abstract as well as in the last paragraph somewhat misleading. ” (reviewer 3).

Reply: We thank the Reviewer for pinpointing the remaining questions that need to be clarified. We have changed the term “ultrafast” to “subpicosecond” and have responded to the specific comments as described below.

The authors have now written a ten pages long rebuttal indicating many changes (and offering a lot of discussions). Still, I am not fully convinced about the novelty of the results (reviewer 1 and 3). Actually, not much is changed with respect to this issue. Also, the reply addressing the contribution of NMR relaxation is not fully convincing (reviewer 3). For example, I do not agree with the statement (Abstract): “This work thus provides a new basis for studying quadrupolar relaxometry in electrolyte or bioelectrolyte systems.” Again, relaxation rates under extreme narrowing conditions are of not much help regarding the shape of the ACF, no “new guidelines” (Introduction) are provided.

Reply: We would like to highlight again that this work provides the first time quantitative and systematic study of quadrupolar relaxation mechanisms in aqueous electrolytes over a broad range of thermodynamic conditions, and an assessment of common models used in the literature to interpret experiments. This was enabled by the use of both experiments and multiscale simulations, yielding good agreement between the measured and computed relaxation rates.

We have revised the aforementioned statement in the Abstract:

This work thus paves the way to the quantitative understanding of quadrupolar relaxation in electrolyte and bioelectrolyte systems.

We have revised the statement concerning “new guidelines” in the Introduction:

This work thus suggests that the subpicosecond collective dynamics of the liquid primarily drive the quadrupolar relaxation at the sodium ion position, **thereby offering insights into the quadrupolar relaxation mechanisms in electrolyte solutions.**

The authors insist using the term “ultrafast dynamics”. I think the notion “ultrafast” usually does not refer to “picosecond” dynamics.

Reply: To avoid confusion with the notion of “ultrafast dynamics”, we have replaced the term with the actual time scale observed (subpicosecond) both in the title of the article as well as in the abstract and main text.